# Peptide-functionalized membrane camouflage for endogenous H₂S-induced photothermal immunotherapy of orthotopic colorectal cancer

Kai Cheng [1,2,5], Fang Zhang[1,5], Jia-Hua Zou[1,3,4,5], Xiao-Ling Lei[1], Xiao-Ting Xie[1], Yan-Bin Guo[1], Guo-Ping Wang[1], Bo Liu [1], Yuan-Di Zhao [1] ✉, Jiang Xia [2] ✉ & Jin-Xuan Fan [1] ✉

The significant challenges pose by the high recurrence and metastasis rates of colorectal cancer (CRC) persist in its diagnosis and treatment. Activating innate immunity in CRC treatment has the potential to reduce drug resistance and side effects. Here, we develop a biomimetic platform by utilizing anti-microbial peptide-functionalized CRC cell membranes to encapsulate a cobalt-based metal-organic framework (C), hereby called peptide-functionalized camouflage C (PfCC). When injected into tumour-bearing mice, PfCC will degrade under the acidic condition of the tumour microenvironment and release cobalt ions, which react with endogenous H₂S to generate black stellate precipitates with good photothermal properties, recruiting NK cells and mitigates the immunosuppressive tumour-microenvironment. Simultaneously, the degradation of PfCC will release structure-protected anti-microbial peptides, inhibiting harmful bacteria, such as *Desulfovibrio*, and reducing H₂S production. The abovementioned synergistic top-down regulation of H₂S promote the polarization of macrophages and further activates the innate immune response. Moreover, experiments including the convex hull algorithm from AI deep learning of the segment anything model indicate that PfCC exhibits the most effective therapeutic effect compared with the single H₂S-regulated therapeutic modality. Taken together, PfCC represents a potential anti-cancer therapy for CRC with the combined effect of immune-regulation and the regulation of the gut flora.

Changes in modern diet and lifestyle have greatly influenced the incidence and development of colorectal cancer (CRC). As worldwide, the first line of treatment for CRC still remains to be surgical resection of the cancerous lesion combined with radiotherapy and chemotherapy[1,2]. However, because of the existence of tumour heterogeneity, the five-year postoperative recurrence rate for CRC at stages II and III is between 35% and 45%[3]. Tumour-associated inflammation has been included as one of the most common feature of tumours[4]. The degree of inflammatory cell infiltration in the tumour tissue depends on the lesion of inflammatory or primary CRC[5–7]. Studies have shown that 70% of the body's immune cells exist in the intestine, 95% of the body's infections occur in the mucosa, and 25% of

the intestinal mucosa is composed of lymphoid tissue[8–11]. However, macrophages, NK cells, and neutrophils, which are widely distributed in the intestine, are mostly in a state of immunosuppression[12–14], further leading to the occurrence and development of diseases. Accordingly, purposeful regulation of intestinal immune cell functions will help us explore the therapeutic model of intestinal diseases.

$H_2S$, the third most common gas signalling molecule following NO and CO, is widely involved in the regulation of respiratory, digestive, and neurological physiological processes[15–18]. Under normal physiological conditions, the main routes of $H_2S$ production in the intestinal tract are the enzymatic pathway, characterized by cystathionine-β-synthase (CBS) and cystathionine-γ-lyase (CSE), and the non-enzymatic pathway, characterized by sulphate-reducing bacteria[19–22]. When intestinal diseases occur, as leading role, sulphate-reducing bacteria can ferment sulphur-containing amino acids, sulphates, and other substances to produce a large amount of endogenous $H_2S$, the content of which is ~3–4 times than that found in the normal intestinal tissue, and the concentration reaches approximately 0.3–3.4 mM[23–26]. Sulphate-reducing bacteria include *Desulphurium*, *Desulfovibrio*, and *Desulfobulbus*, among others. Furthermore, *Desulfovibrio* accounts for ~66% among them in human colon tissues[27]. $H_2S$ is a double-edged sword in intestinal diseases[28–30]. It has shown that, in endotoxin-induced shock in mice, high concentrations of $H_2S$ exhibit pro-inflammatory activity, also can trigger lipopolysaccharide-induced inflammation in mice, which will promote and recruit inflammatory cell infiltration[31,32]. At physiological concentrations, $H_2S$ can reduce the production of proinflammatory cytokines, chemokines, and enzymes, and inhibit the adhesion of leucocytes to vascular endothelium and the migration of leucocytes to the inflammatory site[33–36]. It is also reported that long-term exposure of intestinal mucosa to $H_2S$ is a major factor contributing to the incidence of CRC[37–39]. Therefore, continuous and stable regulation of $H_2S$ levels to physiological concentrations is an effective treatment strategy for alleviating CRC.

An immunosuppressive intestinal tumour microenvironment is closely associated with tumour-associated inflammation[40–44]. Accordingly, regulating the intestinal inflammatory environment can also improve immunosuppressive tumour microenvironment. Now, antimicrobial peptides, as a substitute for antibiotics, have shown important potential in anti-infection therapy, with broad-spectrum antibacterial activity and multiple immunomodulatory effects, but their stability is poor in vivo or in vitro. Because they are easily degraded, resulting in a rapid decline in activity. Although traditional photothermal therapy can minimize the damage to healthy tissues compared with traditional chemotherapy or radiotherapy due to its strong targeting and local effects. However, technical challenges such as deep penetration and local temperature control still existed.

In this work, antimicrobial peptide-coupled phospholipids are embedded into CRC cell membranes, which encapsulate C to prepare biomimetic nano-platforms with specificity and stability, and called peptide-functionalized camouflage C (PfCC) (Fig. 1). Our findings indicate that the antimicrobial peptide has a pronounced inhibitory effect on the intestinal *Escherichia coli* and *Desulfovibrio*. The increase in the levels of intestinal probiotics and the decrease in the levels of harmful flora reshape the CRC tumour microenvironment and improve the inflammatory microenvironment. Moreover, the reduction of *Desulfovibrio* further changes the bacteria-generation pathway of $H_2S$ in the intestine, maintains $H_2S$ at a low level in the intestine, realizes upstream intestinal flora regulation, and downstream $Co^{2+}$ consumption to reduce the high concentration of $H_2S$ in the intestine. These changes promote $H_2S$ to play a positive feedback regulatory role and inhibit the expression of nuclear transcription faction-κB (NF-κB), facilitating the transformation of tumour-related macrophages from cancer-promoting M2 type to cancer-suppressing M1 type, thus improving the immunosuppressive tumour microenvironment, and finally activating the body's inherent immune response. Furthermore,

the adhesion molecules (e.g., adhesion plaque protein and integrins) on the surface of CRC cell membranes endow this platform with homologous recognition and homing properties, enabling rapid enrichment into the tumour microenvironment and enhancing the endocytosis ability of the platform. Moreover, based on AI deep learning of large models, we evaluate the treatment effect of in situ CRC based on algorithms. The results also show that this biodegradable biomimetic platform possesses distinctive drug delivery capability and high biosafety, achieving the therapeutic effect of precipitation photothermal therapy and the activation of intrinsic immunity for in situ CRC. Accordingly, this nanoplatform will offer an expected research avenues for endogenous $H_2S$-based tumour therapy and metastasis inhibition.

## Results
### Synthesis and characterization of PfCC
In this work, using antimicrobial peptide-embedded CRC cell membrane, we encapsulated a degradable framework to fabricate a stable, simple nanoplatform. Because of "homing effect", this platform could be homologously delivered to CRC tumour sites, and then degraded in the acidic tumour microenvironment (pH=6.5), which promoted the release of $Co^{2+}$ from the cobalt-based framework structure. When $Co^{2+}$ reacted with the high concentration of endogenous $H_2S$ at tumour sites, the expression of TNF-α, IL-6, and other inflammatory factors was down-regulated, which inhibited the activation of NF-κB, regulated the relative abundance of M1 and M2 macrophages, and improved the immunosuppressed tumour microenvironment. Furthermore, the $H_2S$-induced cobalt-based precipitate under NIR had excellent photothermal effect, which further activated the inhibitory NK cells and promoted the intrinsic immune system; meanwhile, the release of antimicrobial peptides had a significant inhibitory effect on some Gram-positive and -negative bacteria, especially *Desulfovibrio* and *Escherichia*, which demonstrably improved the balance of intestinal microbiota, and the decreased *Desulfovibrio* further reduced the pathway of $H_2S$ production, and cooperated with the above downstream pathways to change the negative to positive feedback regulation of $H_2S$ in CRC tumour tissues. So the photothermal effect of this platform could be synergistically combined with reducing inflammation in the treatment of CRC tumours.

The TEM results showed that the cobalt-based MOF had a good ortho-hexahedral structure with a size of about 95 nm, which slightly increased to about 115 nm when the antimicrobial peptide-functionalized CRC cell membranes were modified on its surface (Fig. 2a). The dynamic light scattering results demonstrated that the composite particle's size was about 140 nm (Fig. 2b), which was larger than that of the TEM results due to the hydrated particle size of the former. The average surface charges of cobalt-based metal-organic framework (C), CRC cell membrane camouflage C (MCC), and peptide-functionalized camouflage C (PfCC) were about 18.2, 2.3, and 21.4 mV, respectively (Fig. 2c), owing to the presence of numerous positively charged Co frameworks in the C, negatively charged phosphate groups in the cell membrane, and $NH_2$ in the antimicrobial peptide. The EDS analysis (Fig. 2d) and characteristic element spectra (Fig. 2e) of PfCC showed that obvious Co, N, O, P, and S signals appeared, which were further confirmed by the XPS results (Fig. 2f), where Co 2p 3/2 and Co 2p1/2 were located at 781.08 eV and 798.58 eV, N 1 s at 400.83 eV, S 2p at 165.63 eV (Supplementary Fig. 1), P 2p at 133.45 eV (Fig. 2g), S 2p at 165.63 eV (Supplementary Fig. 1), while Co and N were the characteristic elements in cobalt-based MOF, P and S were located in peptide-modified cell membranes. Furthermore, FT-IR spectra (Fig. 2h) showed that the expansion vibration peak of Co-N, plane bending, and expansion vibration peaks of imidazole rings appeared at 420, 500–1350, and 1350–1500 $cm^{-1}$, respectively. The antimicrobial peptide and cell membrane had a carbon-oxygen double bond at 1715 $cm^{-1}$, and the characteristic carbon-sulphur double bond peak at 1030 $cm^{-1}$,

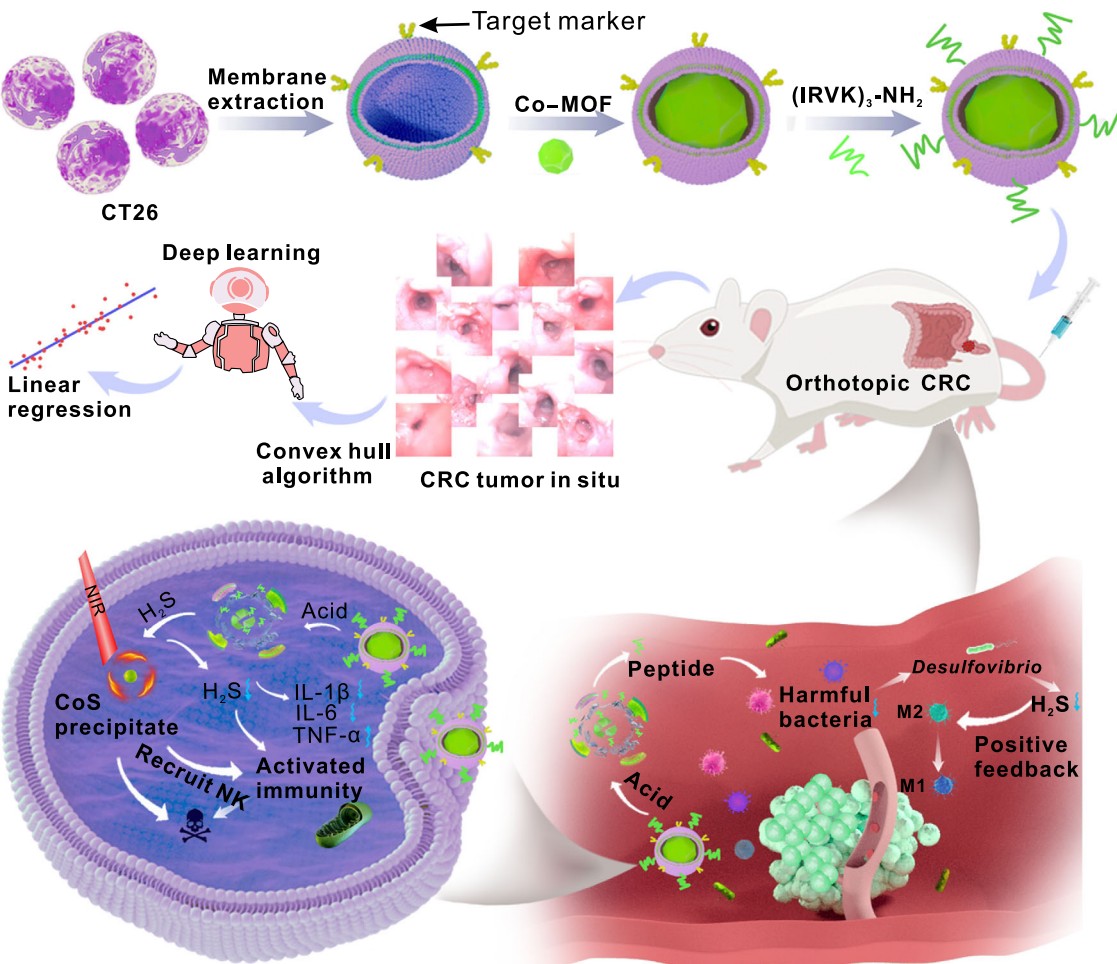

**Fig. 1 | Anti-tumour mechanism of PfCC. 1.** The main steps were as follows: 1. PfCC was synthesized. under the acidic tumour microenvironment (pH 6.5), the cobalt-based platform could degrade and release free $Co^{2+}$ to react with $H_2S$, which initially reduced the $H_2S$ levels in the tumour, and alleviated the occurrence of intestinal inflammation. **2.** PfCC actively targeted orthotopic CRC, and with the reaction of in situ $H_2S$, a precipitate with colorectal photothermal effect was generated to accurately killing tumour cells. **3.** The antimicrobial peptide, modified on the CRC cell membrane, could effectively be released through the disintegration of the cobalt-based structure, and synergistically regulated intestinal flora with reducing the harm of *Desulfovibrio* and other bacteria, and then resulting in promoting the growth of beneficial bacteria. **4.** Based on the large model algorithm, the size and number of CRC tumours were quantitatively analysed, and the real-time dynamic quantification of the therapeutic effect of PfCC also could be realized. The 3D element of Fig. 1 was created with 3D Max, and assembled with CorelDraw.

all the above results indicated the successful modification of the peptide-functionalized cell membrane on cobalt-based MOF surface. The UV-Vis absorption spectroscopy results (Fig. 2i) revealed that the characteristic absorption peak of C was at 590 nm. After loading the fluorescent dye Cy5.5 into the MOF, its peak appeared at 680 nm, and the characteristic peak of Cy5.5 still existed after the modification of the simple cell membrane and the peptide-functionalized cell membrane. The membrane protein content in the PfCC was determined by a standardized BCA detection method (Supplementary Fig. 2), and the results showed that 1 mg of C contained 0.2 mg of membrane protein, laying the foundation for good protein homology targeting probe. Additionally, the content of antimicrobial peptide and Co in PfCC were quantified. One millilitre of PfCC were dried by freeze-drying, and weighed to obtain 4.75 mg of solid powder. With using an atomic absorption spectrophotometer, the standard curve for cobalt was plotted (Supplementary Fig. 3a), and the cobalt ions were quantified in PfCC. The results showed that the concentration of cobalt ions was 0.8076 mg/mL, accounting for 17.00% of the PfCC probe. BCA protein concentration analysis revealed that, after subtracting the protein content in the cell membrane, the protein content of antimicrobial peptide was 0.152 mg/mL, accounting for 3.20% of the PfCC probe (Supplementary Fig. 3b).

Due to the acid-responsive property of cobalt-based MOFs, the degradation of PfCC was investigated at pH 6.5. It was found that PfCC was partially dissociated after 0.1 h (Fig. 2j) and the degradation process was accompanied by the formation of blue precipitate, meanwhile the colour became lighter (Fig. 2j, inset). As the reaction time expanded, the degradation enhanced. After 2 h it was completely degraded with the solution becoming yellowish, and the blue precipitate finally disappeared, because the cobalt-based MOF would agglomerate during the partial dissociation process. $Co^{2+}$ was completely dissociated and dissolved in solution. Then, to simulate the highly concentrated $H_2S$ environment in CRC, different concentrations (1.5, 1.6, 2.0, and 2.1 mM) $H_2S$ standard solution were added to the above pH 6.5 solution after 2 h reaction. The results showed that black precipitate formed continuously as the $H_2S$ concentration increased (Fig. 2k), and the TEM results showed that the addition of $H_2S$ produced numerous aggregate, indicating that the free $Co^{2+}$ resulted from degradation could better consume $H_2S$. Four aliquots of 1 mg/mL MCC were placed in PBS at pH 6.5 for 0.1, 0.5, 1, and 2 h, and after centrifugation, the supernatant were collected for detection (Supplementary Fig. 4). With the quantitative analysis using atomic absorption, the results showed that after 0.1 h, the concentration of cobalt ions in the supernatant was 17.8 µg/mL, and 2 h later, the concentration reached 151.2 µg/mL,

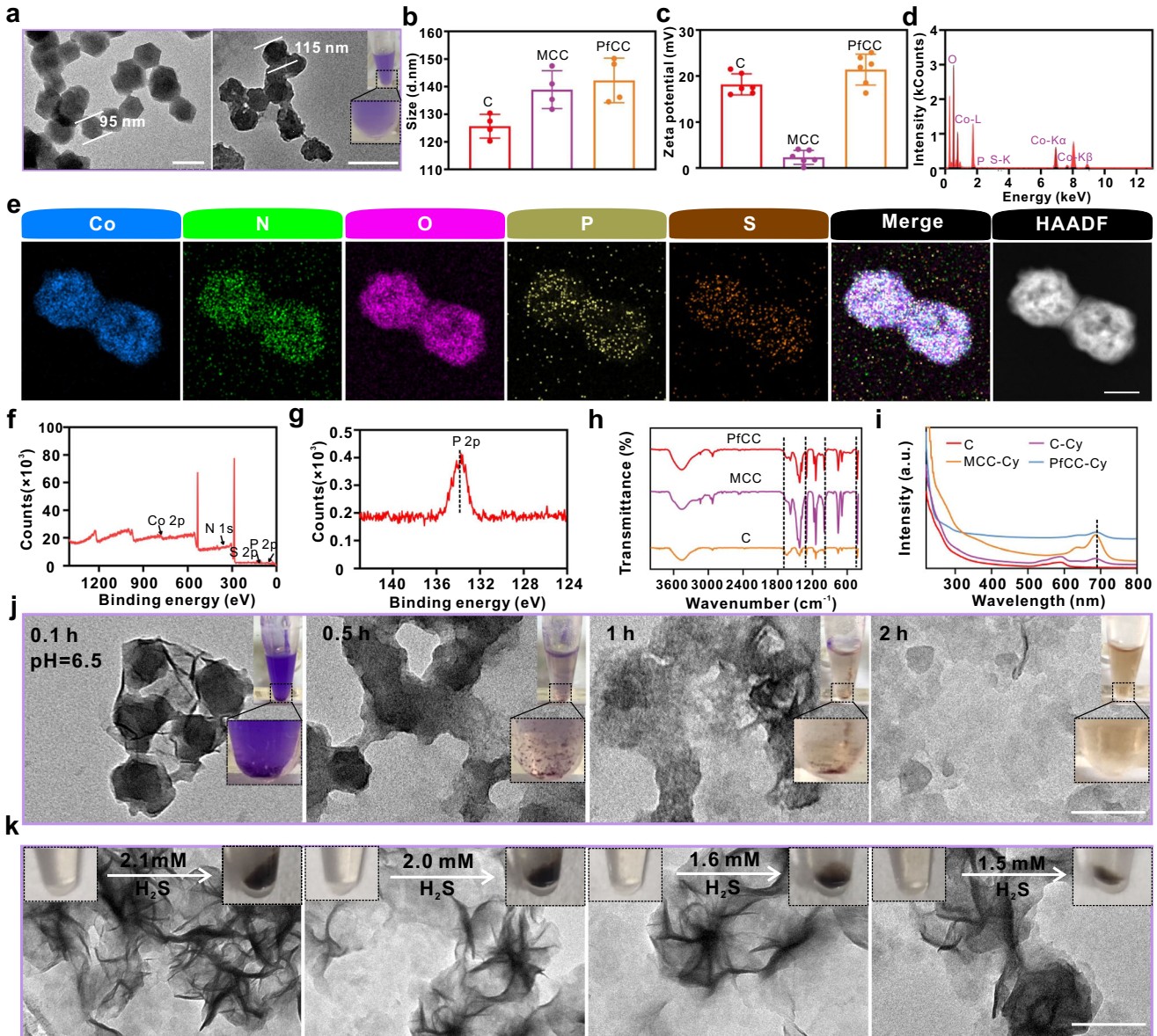

**Fig. 2 | Characterization of the physicochemical properties of the probes. a** TEM of C and PfCC nanoplatforms. Scale bar: 100 and 200 nm from left to right. **b**, **c** Hydrated particle size and zeta potential of C, MCC, and PfCC. Data are presented as mean ± standard deviation ($n = 4$ independent samples for (**b**), and $n = 6$ for (**c**, **d**, **e**) EDS energy spectrum and mapping analysis of PfCC. Scale bar: 100 nm. **f**, **g** X-ray photoelectron spectra of PfCC and high-resolution spectra of element P. **h** Infra-red spectra of C, MCC, and PfCC. **i** UV-vis absorption spectra of C, C-Cy, MCC-Cy, and PfCC-Cy. **j** TEM of PfCC degraded at different times under pH=6.5, inset was a photo of the object. Images are representative of three independent experimental replicates. Scale bar: 300 nm. **k** Precipitate induced by degrading $Co^{2+}$ with different $H_2S$ concentrations, inset were photos of the objects. Images are representative of three independent experimental replicates. Scale bar: 300 nm.

which showed a significant difference compared to that of 0.1 h, meaning the complete degradation of Co-MOF in MCC, and also providing good evidence for the reduction of endogenous $H_2S$ in vivo.

We also performed X-ray diffraction (XRD) characterization of Co-MOF and the generated CoS. The results (Supplementary Fig. 5a) showed that the obvious diffraction peaks of Co-MOF at $2\theta = 7.23°$, $10.38°, 12.7°, 14.78°, 16.5°, 18.08°, 22.13°, 24.39°$, and $26.69°$correspond to the crystal faces (011), (002), (112), (022), (013), (222), (114), (233), and (044), of ZIF-67 (PDF#43–0144). In the case of CoS (Supplementary Fig. 5b), the relatively intense peaks at $2\theta = 31.59°, 35.52°, 44.36°$, and $55.31°$ correspond to the (002), (020), (200), and (120) lattice planes of the structure of CoS respectively; this observation is consistent with the standard card for CoS (PDF#65–3418).

Next, the photothermal property of the above-mentioned CoS precipitate was evaluated. The results showed that under 808 nm near-

infra-red irradiation, the photothermal effect of gradually CoS increased with the increase in precipitate (Fig. 3a). At an $H_2S$ concentration of 2.1 mM and a laser power density of 1 W/cm², the temperature of the CoS precipitate increased by 22.5 °C within 5 min (Fig. 3b), which was enough to achieve the therapeutic killing effect of tumour cells in vivo. We also continued to conduct infra-red thermal imaging studies on different treatment groups, and it was found that the temperature of the PBS, $H_2S$, and $H_2S$ + PfCC groups remained basically unchanged without NIR irradiation. After adding 2.1 mM of $H_2S$ to PfCC and irradiating with NIR laser, the temperature could change by about 20 °C within 5 min (Supplementary Fig. 6).

Good stability was one of the necessary requirements for probe applications. Therefore, in vitro stability simulation experiments were carried out on PfCC. At 37 °C, PfCC probes were mixed with PBS, DMEM, and DMEM + 50% foetal bovine serum. The results showed that

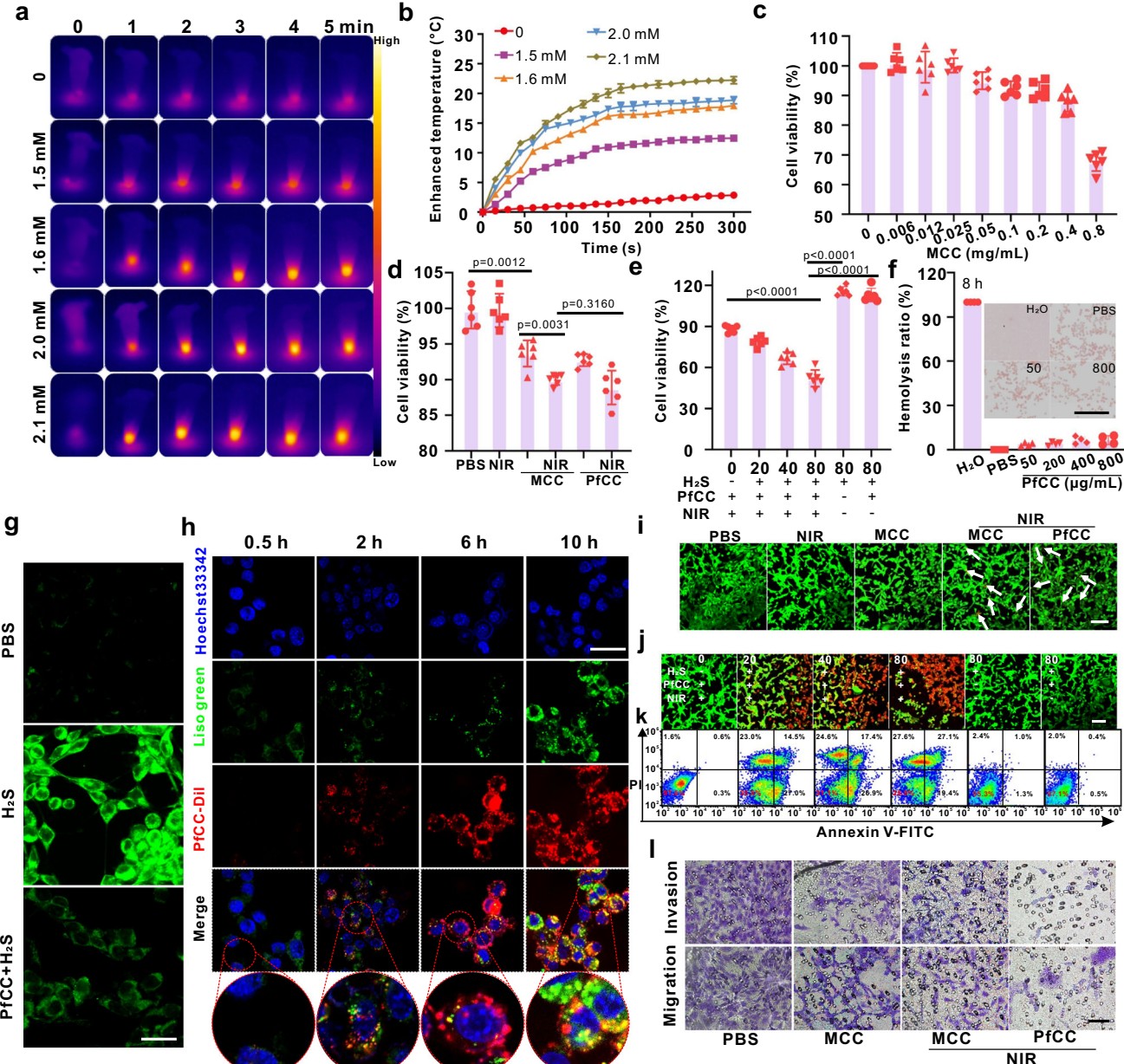

**Fig. 3 | Assessment of probe targeting and therapeutic efficacy at the cellular level. a, b** Thermal and temperature change images of degraded $Co^{2+}$ treated with different concentrations of $H_2S$ standard solution for 2 h followed by laser treatment at $1.0\,W/cm^2$. Data are presented as mean ± standard deviation ($n = 3$ independent samples). Black to white represents the temperature intensity values from low to high. **c** CCK-8 cytotoxicity of CT26 cells incubated with different concentrations of MCC. Data are presented as mean ± standard deviation ($n = 6$ independent samples). **d** Survival rates of CT26 cells incubated with different probes and treatments. Data are presented as mean ± standard deviation ($n = 6$ independent samples). Statistical differences were calculated using two-tailed Student's t test, *: $p < 0.05$; **: $p < 0.01$; ***: $p < 0.001$. **e** Survival rates of CT26 cells after incubation with PfCC treated with different concentrations of $H_2S$ standard solution and irradiation with NIR. Data are presented as mean ± standard deviation ($n = 6$ independent samples). Statistical differences were calculated using two-tailed Student's t test, *: $p < 0.05$; **: $p < 0.01$; ***: $p < 0.001$. **f** Haemolysis rates of erythrocytes incubated with different probe concentrations. The inset showed Giselm staining image of cell smears prepared from the precipitate. Data are presented as mean ± standard deviation ($n = 4$ independent samples). Scale bar: 100 μm. **g** WSP-1 staining of CT26 cells after treatment with different probes. Images are representative of three independent experimental replicates. Scale bar: 40 μm. **h** Co-localization images of Dil-labelled PfCC incubated with CT26 cells for different time. Images are representative of three independent experimental replicates. Scale bar: 40 μm. **i** Fluorescence imaging of CT26 cells stained with calcein and PI after co-incubation with different probes and conditions. Images are representative of three independent experimental replicates. Scale bar: 100 μm. **j, k** Calcein/PI staining and flow cytometric analysis of apoptosis in CT26 cells incubated with different concentrations of $H_2S$-treated MCC. Images are representative of three independent experimental replicates. Scale bar: 100 μm. **l** Transwell experiments of various probes co-incubated with CT26 cells. Images are representative of three independent experimental replicates. Scale bar: 40 μm.

with increasing time, the dispersion PDI was basically stable (Supplementary Fig. 7), the change in zeta potential was small (Supplementary Fig. 8), and the particle size was not statistically different (Supplementary Fig. 9), suggesting that this multifunctional probe had

excellent stability, which laid the foundation for the application of it. To ensure the biosafety of the PfCC probe, the cytotoxicity of MCC was examined using the standard CCK-8 assay. After 24 h of incubation with CT26 (Fig. 3c) and 4T1 (Supplementary Fig. 10) cells, it showed

that the survival rates of CT26 and 4T1 cells remained high (87.8 and 88.4%, respectively) at a probe concentration of up to 400 µg/mL, indicating that the probe had the characteristics of low toxicity. As we know, some antimicrobial peptides can directly inhibit tumour growth. Therefore, we also conducted experiments to verify the cytotoxicity of our used antimicrobial peptides. The results showed that when CT26 tumour cells were incubated with antimicrobial peptides for 24 h, the cell viability did not decrease significantly with the increase of antimicrobial peptide concentration, and when the antimicrobial peptide concentration reached up to 1.28 mg/mL, the cell viability was still above 90% (Supplementary Fig. 11a), while for 48 h incubation with CT26 cells, the cytotoxicity still did not decrease significantly (Supplementary Fig. 11b), indicating that the antimicrobial peptides had no significant toxicity to CT26 cells within 48 h. In addition, the photothermal effect of probe precipitation on CT26 cells was investigated by the CCK-8 method. The results showed that the survival rate of cells with NIR treatment alone was about 98.5% (Fig. 3d), which was similar to that of the PBS control group, indicating that NIR alone had no effect on cell survival status. After co-incubation of MCC with CT26 cells, the cell survival rate was significantly different from that of the PBS group ($p < 0.01$), which might be due to the presence of a small amount of intracellular $H_2S$ that prompted the probe produce CoS precipitate and influence cell growth status. The survival rate of the probe still reached 93.6% at a cobalt-based MOF concentration of 60 µg/mL. When the cells incubated with MCC were treated with NIR, the cell survival rate decreased to 90.0%, which was remarkably different from that of the MCC group ($p < 0.01$), probably due to the small amount of CoS precipitate was generated within the cells, and its weak photothermal effect resulted in a small amount of cell death. After incubation with the PfCC probe and laser irradiation, the survival rate of CT26 cells was 88.8%, and there was no significant difference compared to the peptide-free probe group, indicating that the antimicrobial peptide did not cause significant toxic side effects on CT26 cells in the short term.

To explore the effect of $H_2S$ on the photothermal effect of the probe, different concentrations of $H_2S$ were added to the cells during probe incubation, and $H_2S$ and $H_2S + PfCC$ without NIR groups were used as controls. The results showed that with increasing $H_2S$ concentration, the survival rate of CT26 cells gradually decreased (Fig. 3e), which was significantly different from that of the control group ($p < 0.001$), and the survival rate was only 50.2% when the $H_2S$ concentration reached 80 µM. Since the low concentration of $H_2S$ (230 µM) promotes cell growth and the high concentration (394 µM) inhibits cell growth[45], $H_2S$ had no inhibitory effect on cell viability under these experimental conditions, so we could conclude that the decreasing cell survival rate of CT26 cells was due to the reaction of $H_2S$ with $Co^{2+}$ to form a CoS precipitate, which produced a thermal effect and killed cells under NIR irradiation. Subsequently, we studied the effect of PfCC on intracellular $H_2S$ using the WSP-1 probe, and the results demonstrated that a small amount of green fluorescence appeared in the control group (Fig. 3g), indicating that there was some dissolved $H_2S$ in the cells; after the addition of the $H_2S$ solution, a clear fluorescence was observed in CT26 cells, and after incubation with PfCC, the amount of intracellular green fluorescence decreased significantly, which might be attributed to the high $H_2S$ consumption by $Co^{2+}$ after degradation, further indicating that $Co^{2+}$ could effectively consume $H_2S$ gas in tumour cells. Subsequently, we conducted the photothermal conversion efficiency testing experiments of CoS (Supplementary Fig. 12). Since CoS is poorly soluble in water, 1 mg of CoS powder was added to 1 mL of DMSO containing 0.01 mg of sodium dodecyl sulphate. After ultrasonication for 30 mins, the UV-vis absorption spectrum absorption of CoS at 808 nm was measured as 1.31. The 300 µL of CoS was placed in a quartz cuvette and irradiated with an 808 nm laser for 5 min, then recording the temperature of

heating and cooling processes. According to the formula (1),

$$\eta = \frac{hs(T_{Max} - T_{Surr}) - Q_{Dis}}{I(1 - 10^{-A_{808}})} \tag{1}$$

here, $T_{max} = 53.5$, $T_{surr} = 26.1$, $A_{808} = 1.31$, the power intensity is 600 mW, and h=mc/τ, t=τ (·LnΘ), the photothermal conversion efficiency could be calculated to be 20.72%.

Based on the killing effect of the above probe on CT26 cells, the intracellular internalization process of PfCC was further investigated. Here, confocal laser scanning microscopy was adopted to examine CT26 cells incubated with PfCC, in which lysosomes were labelled with Lyso-Tracker green to show green fluorescence, PfCC was labelled with DiI to show red fluorescence, and the nuclei were labelled with Hoechst 33342 to show blue fluorescence. The results showed that with increasing PfCC incubation concentration, the red and green fluorescence showed a gradually increasing overlap after 4 h of incubation with CT26 cells (Supplementary Fig. 13), which was located in the cytoplasm, indicating that the DiI-PfCC could be phagocytosed by CT26 cells.

Since PfCC entering the cells was initially phagocytosed by lysosomes, whether PfCC had the lysosomal escape ability was further investigated. Lysosomal colocalization experiments were performed by incubating CT26 cells with DiI-PfCC for different time (0.5, 1, 2, 4, 6, 8, and 10 h). The results showed after 0.5 h incubation (Fig. 3h), the intracellular red fluorescence was weak, which might be due to the lower phagocytosis of the DiI-PfCC probe by CT26 cells. With the extended incubation time to 4 h, the red fluorescence of DiI-PfCC largely overlapped with the green fluorescence of lysosomes (Supplementary Fig. 14), but overlapped less as the incubation time was further extended, and after 10 h, the overlapped fluorescence became basically separated, which further indicated that with increasing incubation time, DiI-PfCC achieved lysosomal escape, and the above results were also confirmed by the quantitative analysis of fluorescence colocalization (Supplementary Fig. 15), the red fluorescence of DiI-PfCC and the green fluorescence of Lyso-Tracker green on lysosomes gradually overlapped and separated over time.

Based on the good phagocytosis effect of CT26 cells, the synergistic therapeutic effect of PfCC was further verified, and confocal imaging experiments with calcein-AM and PI double staining were performed on cells after different treatments. Calcein-AM can penetrate the cell membrane of living cells and hydrolyse into calcein by intracellular esterases to emit green fluorescence, so it can label living cells; PI can emit red fluorescence after intercalation of double-stranded DNA, but cannot pass through intact cells with cell membranes, so dead cells can be labelled. The results showed that CT26 cells exhibited strong green fluorescence after PBS treatment with/without laser irradiation (Fig. 3i), indicating that laser treatment alone could not cause massive cell death. The $H_2S$ alone and the $H_2S + PfCC$ group both exhibited green fluorescence, indicating that the apoptotic effect on cells caused by low concentrations of $H_2S$ and the simple CoS precipitate was relatively small. After the cells were incubated with MCC and irradiated with NIR, little red fluorescence appeared, indicating that the probe produced a certain thermal effect. Compared with the MCC + NIR group, no more similar red fluorescence was observed in the CT26 cells after PfCC+NIR treatment, indicating that the antimicrobial peptide had no significant toxic side effects on CT26 cells. When $H_2S$ solution was added, red fluorescence appeared in CT26 cells and gradually increased with the increasing concentration (Fig. 3j), which might be due to the degradation of the targeted C, leading to the release of numerous Co ions to form CoS precipitate, which resulted in the enhancement of the photothermal effect of the precipitate to produce apoptosis. But the $H_2S$ alone and the $H_2S + PfCC$ group both exhibited green fluorescence, indicating that the apoptotic

effect on cells caused by low concentrations of H₂S and the simple CoS precipitate was relatively small.

For quantitatively investigating the apoptosis induced by MCC with different H₂S concentrations, FITC-PI flow cytometry staining analysis of CT26 cells was performed. The results showed when no H₂S was added (Fig. 3k), only a few late-stage apoptotic cells (1.6%) were present in this system; when simple H₂S was added, only a few late-stage apoptotic cells (2.4%) were present; after adding PfCC, late-stage apoptotic cells changed less, which indicated that the simple H₂S and formation of CoS precipitates generally didn't cause any damage to the CT26 cells. When low concentration of H₂S was added, the mild photothermal heating induced by H₂S was not enough to cause large-area necrosis of cells (Q1), but more early apoptosis (Q3) and late apoptosis (Q2). As the concentration of H₂S increased, the degree of late apoptosis became too severe, the probability of necrosis may gradually increase, which also showed that continuous local hyperthermia will cause increased cell necrosis. From the Fig. 3k, when CT26 cells incubated with MCC were added with 20 μg/mL of H₂S and treated with laser irradiation, the early and late apoptosis were 27.0 and 14.5%, respectively, and the cell necrosis reached 23%. But when the H₂S concentration was 80 μg/mL, the late-stage apoptosis and necrosis reached 27.1% and 27.6%, and the total percentages of the early stage apoptosis, later stage apoptosis and necrosis were 64.5%, 68.9%, and 74.1% respectively, indicating that high concentrations of H₂S may induce a stronger photothermal effect of $Co^{2+}$ generated by acid degradation in CT26 cells. This was essentially in line with the trend of the above-mentioned confocal imaging experiments, which laid a good foundation for the in vivo precipitation of photothermal therapy.

To verify the ability of the probe to inhibit the migration and invasion of CT26 cells, a transwell experiment was performed (Fig. 3l). It was found that stained CT26 cells appeared more, the frequency of migration and invasion was higher in PBS group; whereas CT26 cells treated with PfCC appeared less, the migration and invasion of CT26 cells after NIR irradiation was further reduced, which might be due to the apoptosis of a small number of cells under the precipitation photothermal effect; antimicrobial peptides conjugated probe further reduced the migration and invasion of CT26 cell, owing to the fact that antimicrobial peptides had the ability to inhibit cell migration and invasion to some extent, which was also confirmed by cell counts (Supplementary Fig. 16). The above results demonstrated that this probe had the ability to inhibit the metastasis of CT26 cells, and laid the foundation for subsequent metastasis inhibition in vivo.

Since the probe entered the body via the blood circulation and was subsequently distributed throughout the body, the hemocompatibility of the probe was investigated. The results showed that the haemolysis rate of erythrocytes was 100% in ultrapure water (Fig. 3f) and 0% in PBS, which was attributed to the inequality of osmotic pressure. When the probe concentration reached 0.8 mg/mL and incubated for 8 h, the haemolysis rate remained low, and Giemsa staining microscopy of the cell smear showed that the cell morphology was still regular with bright red colour (Fig. 3f, inset), which was basically similar to the PBS group, while the cells incubated in pure water were in fragmented state with bright red background. These results indicated that PfCC exhibited good compatibility with blood cells and stability in blood, and had minimal effect on red blood cells, which provided basic safety for subsequent in vivo experiments.

### Biocompatibility of PfCC in vivo

Given the positive experimental results in vitro, and to investigate whether the probe had any potential toxicity in vivo, the biochemical index in blood was examined. The results showed that white blood cell (WBC), red blood cell (RBC), haemoglobin (HGB), and platelet (PLT) counts in healthy BALB/c mice injected with PfCC for 1, 20, and 45 d were not significantly different from those of the PBS group (Supplementary Fig. 17a–d); furthermore, ALT and AST levels were within the

normal range (Supplementary Fig. 17e and f), suggesting that the probe had no significant impact on liver and kidney function in mice. The heart, liver, spleen, lung, and kidney index (Supplementary Fig. 17g–k) of mice after necropsy were also in the normal range compared with those of the PBS group. In addition, the change in mouse weight was also one of the indicators of mouse health, the influence of PfCC on mice was also measured by monitoring the change of body weight after probe injection. The results showed that the body weight of mice was not significantly different from that of the control group (Supplementary Fig. 17l), and there was no notable weight loss as all mice remained within the normal range. The above findings further supported that PfCC exhibited favourable low toxicity and biosafety in vivo. The HE results revealed that after PfCC injection, the morphology and structure of all tissues in mice were normal and clear after 1, 20, and 45 d (Supplementary Fig. 17m). Here, the transverse stripe of the heart was clearly visible, the hepatic lobules and other characteristic structures of the liver were obvious without oedema, the splenic corpuscle of the spleen was apparent with the central artery, clear alveoli were observed in the lungs with no thickening of the alveolar wall, the glomeruli and surrounding tubules in the kidneys were clearly distributed, and no swelling of the villi and obvious macrophage infiltration were present in the small intestine, which indicated that the organs and tissues were basically normal, and PfCC would not cause pathological lesions. In addition, an in vivo metabolism study of PfCC probe was conducted in CT26 tumour-bearing mice at different time (Supplementary Fig. 18a and b). The results showed that after 24 h injection of PfCC probe, the metabolic elimination rate of cobalt ions in the blood reached 99.9%; 99.8% in the heart; 27.1% in the liver, and 91.5% after 7 d; 45.3% in the spleen, and 89.0% after 7 d; 87.9% in the lung, and 99.9% after 7 d; 88.0% in the kidney, and 100% after 7 d; 31.3% in the tumour, and 97.7% after 7 d. The aforementioned experiments confirmed that PfCC had good biocompatibility and high long-term in vivo safety, establishing a solid foundation for the deployment of the probe in vivo.

### Imaging of PfCC in vivo

Based on the positive results in vitro and satisfactory safety tests in vivo, in vivo application of the probe was carried out. Here, Cy5.5-labelled PfCC was injected into CT26 tumour-bearing BALB/c mice through the tail vein, and the overall fluorescence imaging of the mice was continuously monitored. The results showed that after 2 h of injection, the fluorescence signal appeared at the tumour site (Fig. 4a and Supplementary Fig. 19), and gradually increased with time, reaching the maximum at 6 h and then gradually decreasing. Furthermore, the signal-to-noise ratio of the tumour initially increased and then decreased (Fig. 4b, inset), indicating that the probe had achieved effective and continuous enrichment at the tumour site. When the probe was injected into BALB/c mice bearing CT26 and 4T1 tumours on the left and right sides, respectively (Fig. 4a and Supplementary Fig. 20), the fluorescence of PfCC was observed to be consistently stronger in CT26 tumours than that of 4T1 tumours at the same time (Fig. 4c). The tumour fluorescence ratio reached its maximum at 24 h, and subsequently decreased with time, indicating that PfCC could exhibit the enhanced targeting and enrichment capabilities for CT26 tumours.

In this study, BALB/c mice subcutaneously inoculated with luciferase-labelled CT26 cells (Luc-CT26) were intended to examine the probe distribution in organs by tail vein injection. The results of the blood test showed that the blood half-life of PfCC was $3.16 \pm 0.08$ h (Fig. 4d), indicating that the probe displayed excellent blood circulation ability, which laid the foundation for the enrichment of the probe at the tumour site. Fluorescence analysis of tumours and organs removed at different time, along with the bioluminescence imaging of Luc to localize the tumour site, also confirmed the aforementioned fluorescence change process in vivo. After 6 h, the fluorescence signal

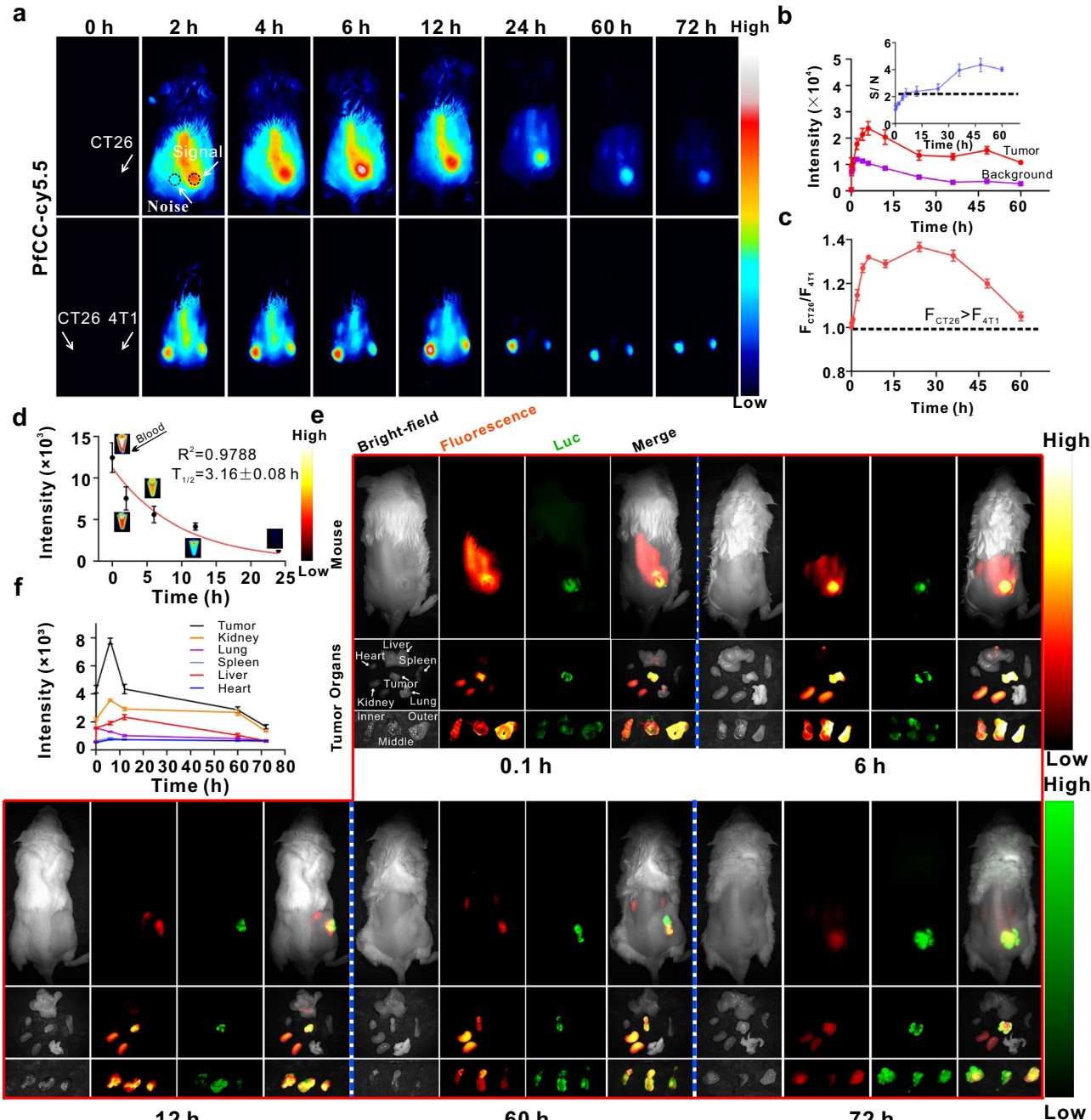

**Fig. 4 | Imaging of subcutaneous tumours with the probe. a** Fluorescence imaging of unilateral subcutaneous CT26, bilateral subcutaneous CT26 and 4T1 tumours at different time after injection of probes into the tail vein. Black to white represents the fluorescence intensity values of Cy5.5-labelled PfCC from low to high. **b** Fluorescence intensity values and signal-to-noise ratio of unilateral tumour versus background region. Data are presented as mean ± standard deviation ($n = 3$ independent mouse). Black to white represents the fluorescence intensity values of Cy5.5-labelled PfCC from low to high, and for tumour autofluorescence, black to green represents the fluorescence intensity values of Luc-CT26 cells from low to high. **c** Signal ratio of bilateral tumour. Data are presented as mean ± standard deviation ($n = 3$ independent mouse). F the abbreviation of fluorescence. **d** Blood half-life. Data are presented as mean ± standard deviation ($n = 3$ independent mouse). **e, f** Fluorescence and bioluminescence imaging of major organs and tumours, and fluorescence intensity analysis of corresponding organs. Data are presented as mean ± standard deviation ($n = 3$ independent mouse).

value of the tumour site was markedly stronger than that of other time (Fig. 4f), which was consistent with the overall results of fluorescence imaging of mice. Meanwhile, the signal in the kidneys was always in a strong state (Fig. 4e), because the probe was metabolized more by the kidney. Additionally, the division of the tumour into three slices from outside to inside revealed a gradual increase in fluorescence intensity over time (Fig. 4e, and Supplementary Fig. 21), indicating that the probe penetrated deeper into the tumour, which provided further

evidence that PfCC had an effective targeting effect inside the tumour. The above experiments demonstrated that the targeted PfCC could be accurately targeted to the CRC tumour site via the blood circulation.

**Anti-tumour targeting for subcutaneous CRC tumours in vivo**
Subsequently, based on the above positive overall fluorescence imaging results, the immediate inhibitory effect on tumour growth was further proceeded. The TUNEL and HE staining of tumour tissue

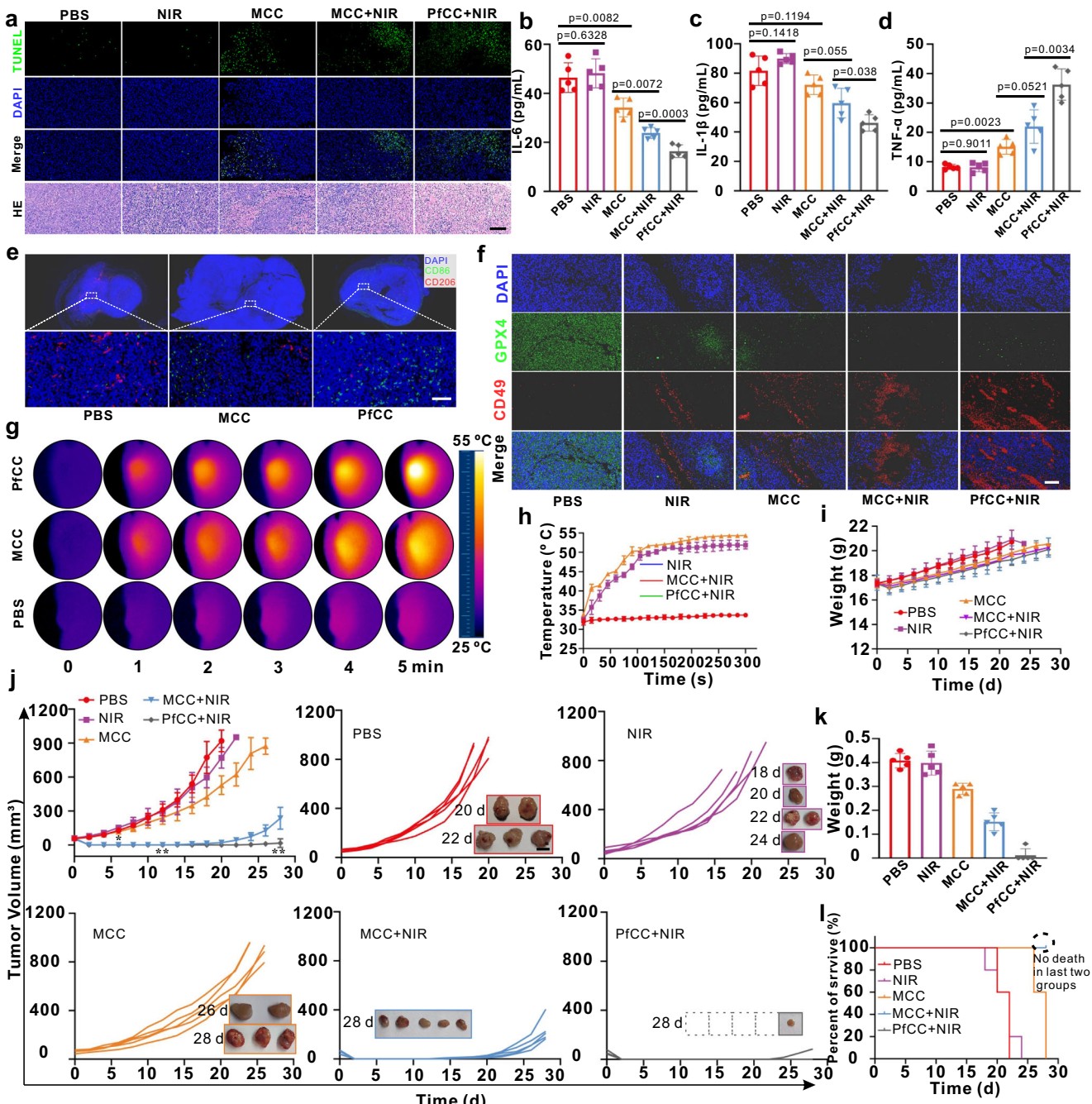

**Fig. 5 | Antitumour study for subcutaneous CRC. a** TUNEL and HE staining of tumour sites in subcutaneous CT26 tumour bearing mice treated with PBS, NIR, MCC, MCC + NIR, and PfCC+NIR. Images are representative of three biologically independent mice. Scale bar: 200 μm. **b–f** ELISA of IL-6, TNF-α, and IL-1β, and Immunofluorescence staining of CD86 and CD206, CD49 and GPX4. Data are presented as mean ± standard deviation (n = 5 independent mouse for (**b**, **c**, and **d**). Statistical differences were calculated using two-tailed Student's t test, \*: p < 0.05; \*\*: p < 0.01; \*\*\*: p < 0.001. Images are representative of three biologically independent mice for e and f. Scale bar: 200 μm. **g**, **h** Thermograms of tumour temperature over time after NIR treatment with PBS, MCC, and PfCC, and their temperature change curves. The data are presented as the mean ± standard deviation (n = 3 independent mouse for h). Black to white represents the temperature intensity values of tumour site from low to high. **i–l** Body weight changes, tumour volume, tumour weight, and survival curves of mice after different treatments. The data are presented as mean ± standard deviation (n = 5 independent mouse). Scale bar: 1 cm.

sections showed that tumour cell growth was unaffected in the PBS and laser irradiation groups alone, and there was no significant cell necrosis (Fig. 5a). However, a moderate degree of necrosis was observed in the MCC group. This might be attributed to the fact that the presence of abundant dissolved H₂S at the inflammatory tumour site was consumed by Co²⁺ after degradation of nanoplatform in an acidic environment, which weakened the inflammatory microenvironment of the tumour site, promoted the conversion of

macrophages from M2 to M1 and strengthened the body's intrinsic immunity. Compared with the above group, the tumour sites in the MCC + NIR group showed different degrees of vacuolazation, cell necrosis, atrophy, and nucleoplasm separation, which indicated that the photothermal effect of CoS precipitation further killed intestinal tumours, coordinated with the alleviation of tumour inflammatory environment, activated innate immune response, and initially achieved the inhibition of the growth of CRC. The most pronounced evidence of

cellular necrosis was observed in the PfCC+NIR group, which might be due to the regulatory effect of the antimicrobial peptide on the bacterial flora in the tumour. The HE staining results also demonstrated that the necrosis of PfCC+NIR cells was most obvious, further supporting the effectiveness of targeted enhancement of the precipitated photothermal effect in conjunction with intrinsic immune activation. The ELISA results of serum inflammatory factors showed that compared with PBS group, there was no significant difference in the expression levels of IL-6 and IL-1β in the NIR group, suggesting that NIR alone would not alter the inflammatory response; however, a significant difference appeared in the MCC group compared with those of the PBS group, with a different degree of decrease (Fig. 5b, c). This might be because the $Co^{2+}$ generated by the degradation of cobalt-based MOFs in tumours consumed intratumoural $H_2S$, thereby attenuating the inflammatory environment in tumours, while IL-6 and IL-1β showed a close positive correlation with M2-type polarization in tumour-associated macrophages, which in turn changed the relative abundance of M1 and M2 macrophages in tumours, and influenced the appearance of innate immunity in tumours. The expression levels of IL-6 and IL-1β in the MCC + NIR group were further reduced, which was significantly different from those in the MCC group, because NIR irradiation further promoted the photothermal effect of CoS precipitation, and the body's inflammation level was gradually relieved. Compared with the MCC + NIR group, because of the positive regulatory effect of antimicrobial peptides on the inflammatory tumour microenvironment, there were also significant differences in the expression levels of IL-6 and IL-1β in the PfCC+NIR group. The expression level of TNF-α was closely related to the tumouricidal effect of the probe. Compared with the PBS group, no notable discrepancy in the expression level of TNF-α in the NIR group, because the temperature change caused by NIR alone was insufficient to kill tumour cells. Significant difference appeared between the MCC and PBS groups in the expression level of TNF-α, which might be due to the down-regulation of $H_2S$ level in tumours by $Co^{2+}$ to promote the expression of M1 macrophages, and to activate the body's innate immune response, thereby achieving the initial elimination of tumour cells. After NIR treatment, the expression level of TNF-α increased, which was attributed to the benifical photothermal effect of CoS precipitate to further kill tumour cells. After modification with an antimicrobial peptide, the expression level of TNF-α also increased due to the regulation of the inflammatory tumour microenvironment by antimicrobial peptides to further activate the body's immune response and achieve the clearance of tumour cells. So because the treatment effect occurred, the level of tumour necrosis factor gradually increased (Fig. 5d).

CD86 and CD206 were highly expressed in M1 and M2 macrophages, respectively. To further verify the regulation of macrophage reprogramming, the red fluorescent dye-labelled CD206 antibody and the green fluorescent dye-labelled CD86 antibody were performed for immunofluorescence labelling of different phenotype macrophages in tumours (Fig. 5e). The results demonstrated that in the PBS group, immunosuppressed M2 macrophages were the predominant phenotype, whereas M1 macrophages increased significantly after MCC treatment, and the abundance of M1 macrophages was the highest when the antimicrobial peptide was conjugated, which confirmed that $Co^{2+}$ and peptides reprogramed the phenotype of tumour-associated macrophages synergistically. Meanwhile, immunostaining of tumours with CD49 (high expression in NK cells, marked in red) and GPX4 (antioxidant systematized core regulatory enzyme, marked in green) were performed, the amount of NK cells was observed to increase significantly in the MCC group (Fig. 5f), and further increased after peptide conjugation and NIR treatment, which was probably associated with the alleviation of immunosuppression in the tumour microenvironment. Additionally, as the expression level of GPX4 increased significantly in tumour tissues compared with normal tissues

and decreased with the alleviation of the oxidative environment, the expression level of GPX4 decreased in the experimental group, further validating the synergistic regulatory effect of the probe in the tumour microenvironment.

Subsequently, the statistical results of the tumour site (Fig. 5g and h) showed that after NIR irradiation, the temperature at the tumour site in the PBS group reached 38.5 °C, which was not enough to kill the tumour cells, whereas it increased to 52.5 and 53.3 °C after 6 h injection of MCC and PfCC into subcutaneous CRC tumour and subsequent irradiation with a low intensity NIR laser (5 min). This showed that the probes were successfully targeted to the tumour site, and produced a notable photothermal effect after laser irradiation, which also meant that they had the ability to trigger apoptosis of tumour cells through photothermal effect.

Given the favourable immediate treatment effect, the long-term effect of subcutaneous tumour treatment was evaluated. From the fluorescence imaging results (Fig. 4a), it was clear that after 4–12 h of injection, the PfCC in the tumour was maximized, so post-injection of the probe into the tail vein at 6 h was chosen for the study as the synergistic treatment. The results showed that after 2 d of probe injection and various treatments, the body weight of the mice in each group decreased to a certain extent (Fig. 5i), which might be due to the poor appetite caused by the effect of the anaesthetic, and the weight increased after 4 d, then changed essentially the same in all groups, indicating that the probe had no apparent toxicity in mice. According to the tumour size (to meet the requirements of animal ethics, the mice were euthanized when the tumour volume reached 1000 mm³), the treatment effect could be roughly divided into the following four categories (Fig. 5j): I) the PBS and NIR groups, the tumour change trends of both groups were similar and the tumour volume increased with time, indicating that PBS and laser alone had almost no therapeutic effect on tumours. II) The MCC group, the tumour growth was significantly suppressed within 6 d, but continued to grow in the following days, and was remarkably differed from category I ($p < 0.05$). This might be attributed to the fact that MCC consumed $H_2S$ at the tumour site, resulting in a temporary improvement in the inflammatory environment, activation of the body's inherent immunity, and suppression of tumour growth to a certain extent. (III) The MCC + NIR group, tumour growth was significantly inhibited within 12 d, and tended to be normal thereafter, but the growth rate slowed down compared with that of category II, and there was a significant difference ($p < 0.01$). These results indicated that on the one hand, the enrichment of targeted probes at tumour sites consumed $H_2S$ to improve the inflammatory environment, regulated the relative abundance of M1 and M2 macrophages, ameliorated the immunosuppressive tumour environment, and activated the body's innate immune response at tumour sites, on the other hand, the tumour cells were killed by the photothermal effect of cobalt sulphide precipitation, thereby synergistically inhibiting tumour growth. IV) The PfCC+NIR group, enhanced photothermal and anti-inflammatory effect was clearly evident, resulting in complete tumour disappearance, and no recurrence until 28 d later. In addition, a significant difference was observed between the tumour size and that of group III ($p < 0.01$), indicating that PfCC fully exerted the effect of endogenous gas-triggered precipitation photothermal effect cooperated with anti-inflammatory therapy. This might be because the antimicrobial peptide had a significant inhibitory effect on Desulfovibrio, which further inhibited the production of $H_2S$ in CRC tumours from the upstream, synergized with $Co^{2+}$ to deplete $H_2S$, effectively attenuated the immunosuppressive microenvironment in tumours, and realized the recruitment and activation of innate immune cells such as M1 macrophages and NK cells under the regulation of microbiota. To verify whether the antimicrobial peptides had an inhibitory effect on tumour growth, we intratumourally injected antimicrobial peptides into CT26 tumour bearing mice. The results showed that during the 20 d

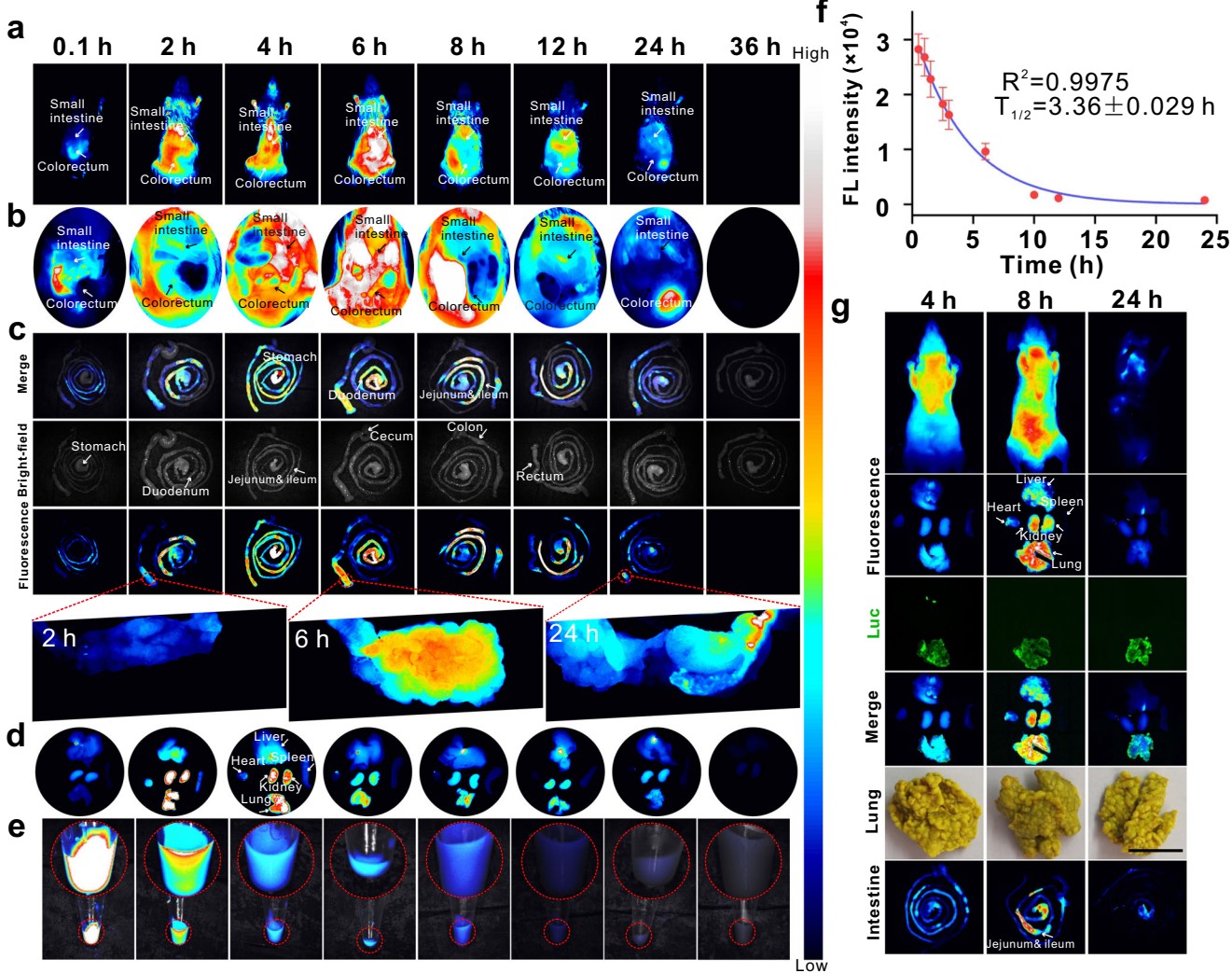

**Fig. 6 | In situ CRC imaging with probes. a** FVB-induced CRC tumours at different time after tail vein injection of probes in mice. Fluorescence imaging of post-dissected **b** abdomen, **c** intestine, **d** organs, and **e** blood. Black to white represents the fluorescence intensity values of Cy5.5-labelled PfCC from low to high. **f** Blood half-life analysis. FL: Fluorescence. The data are presented as mean ± standard deviation ($n = 3$ independent mouse). **g** Fluorescence and bioluminescence imaging of metastatic CRC. Scale bar: 1 cm.

treatment period, the weight of mice increased normally (Supplementary Fig. 22a), and compared with the PBS group, there was no significant difference of tumour volume in the antimicrobial peptide group (Supplementary Fig. 22b and c). HE and TUNEL staining of the tumours also showed that there was no significant tumour cell apoptosis and necrosis in the antimicrobial peptide treatment group (Supplementary Fig. 22d). So the antimicrobial peptides we used had virtually non-existent cellular and in vivo inhibition of growth. After 28 d, tumour weight analysis of different groups also showed that the synergistic treatment effect was the most effective (Fig. 5k). The bright-field images (Fig. 5j, inset) and survival curve analysis (Fig. 5l) of the mice after 28 d or death showed that the mice in PBS group basically died at 20–22 d, NIR group at 18–24 d, and MCC group at 26–28 d. Although, the mice in MCC group did not die, the tumour recurrence rate was 100%. For PfCC+NIR group, the tumour recurrence and survival rate were 20% and 100%, which further demonstrates the excellent long-term therapeutic efficacy of our probe. In conclusion, the photothermal effect of the precipitate combined with inflammation alleviation therapy was markedly better than that of the precipitate photothermal therapy or inflammation alleviation treatment mode. PfCC demonstrated effective photothermal-intrinsic immune

activation combined therapy without notable toxicity in mice, and the low toxicity and high efficacy of the probe provided promising foundation for the development of tumour therapeutic strategies.

Subsequently, therapeutic studies of PfCC were conducted by induced orthotopic CRC tumour models. The initial targeting experiments in vivo (Fig. 6a, b, and Supplementary Fig. 23, 24) demonstrated that after 2 h of probe injection, the fluorescence signals began to accumulate in the intestinal area, and gradually moved from the small intestine to the colon site. The anatomical results of the stomach and the entire intestine (Fig. 6c) demonstrated that after 4–6 h of probe injection, the fluorescence signals in the stomach were highly enriched, and then gradually weakened due to the rapid probe distribution through the gastric artery via the blood circulation. It was also observed that signals in the duodenum increased starting at 2 h and reached a maximum at 6 h, while the jejunum and ileum reached their maximum at 8 h and the colourectum was essentially identical to the duodenum. The anatomical results of the major organs (Fig. 6d, Supplementary Fig. 25) showed that more probes were enriched in the lungs at 2 h, then gradually diminished. Additionally, the fluorescence signal of the kidneys also reached a maximum at 2 h, which might be attributed to the renal metabolism of the probe. The results of blood

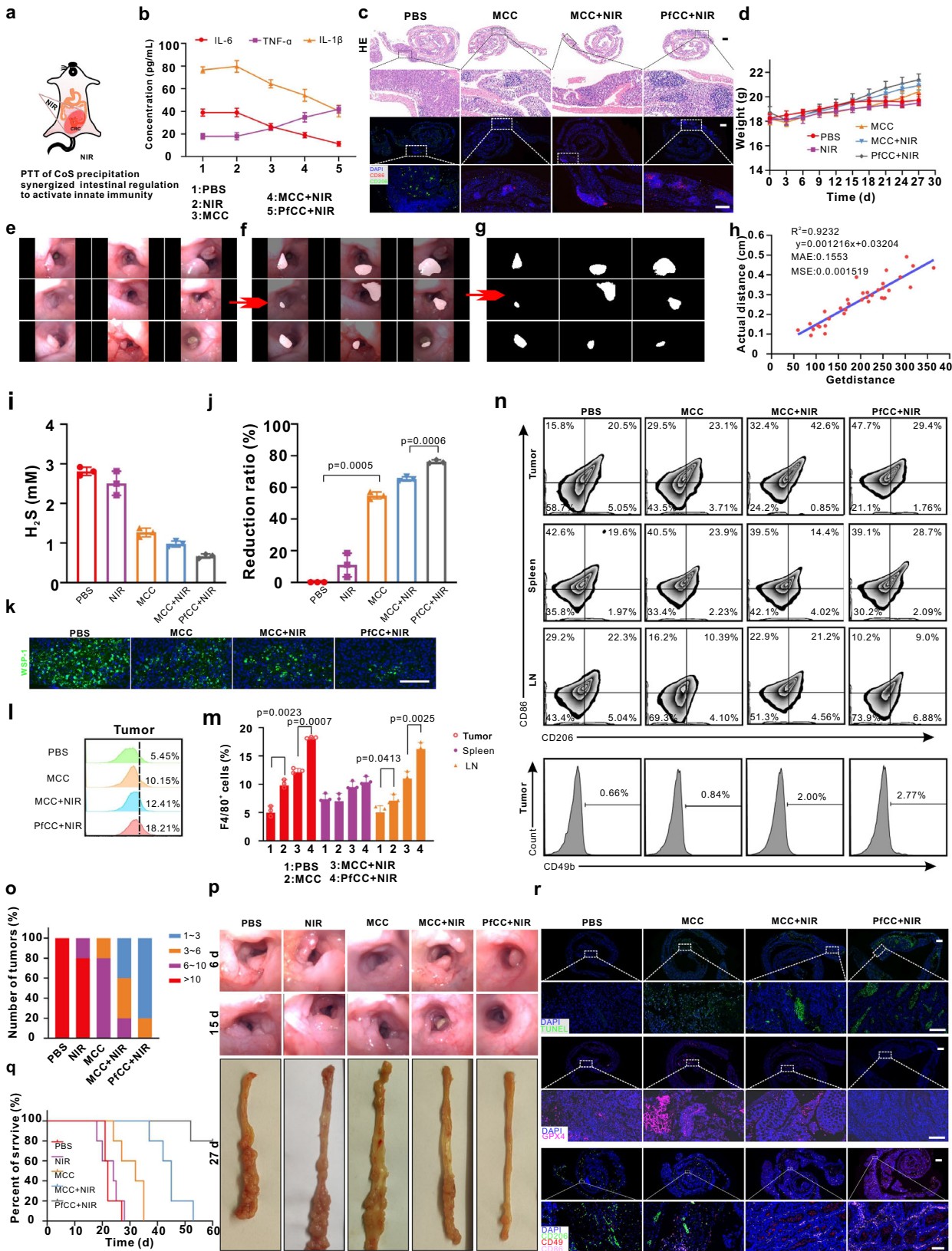

test demonstrated that the half-life of PfCC was $3.36 \pm 0.029$ h (Fig. 6f, e), indicating that it had good blood circulation ability, which laid the foundation for enrichment of probes at tumour sites.

Furthermore, the targeting ability of PfCC in lung metastatic CRC was investigated. It was observed that the lung fluorescence intensity reached its maximum after 8 h of probe injection (Fig. 6g), and

achieved a good overlap with the bioluminescence signal at the tumour site, while in orthotopic CRC tumour (Fig. 6d) reached the maximum enrichment at 4 h, thereby excluding the possibility that the probe enrichment effect from blood circulation was a factor, so the enrichment of probes in this lung metastasis model was determined to be targeted enrichment. After 24 h, only the bioluminescence signals

**Fig. 7 | In situ anti-tumour study of CRC. a** Treatment diagram of in situ induced CRC mouse. **b** IL-6, TNF-α, and IL-1β of ELISA detection after 12 h of PBS, NIR, MCC, MCC + NIR, PfCC+NIR treatment. Data are expressed as mean ± SD ($n = 3$ independent samples). Figure 7a was created with CorelDraw. **c** Immunofluorescence staining of HE, CD86, and CD206. Images are representative of three biologically independent mice. Scale bar: 1 mm, 1 mm, and 200 μm from top to bottom. **d** The weight change after different treatments. Data are expressed as mean ± SD ($n = 5$ independent mouse). **e** Original tumour. **f** Segmented tumour based on the SAM macromodel. **g, h** Extracted tumour and fitting analysis based on pixel vs. actual distance. **i** H₂S content in tumours after different treatments. Data are expressed as mean ± SD ($n = 3$ independent mouse). **j** Reduction rate of different treatment for H₂S. Data are expressed as mean ± SD ($n = 3$ independent mouse). Statistical differences were calculated using two-tailed Student's t test, *: $p < 0.05$; **: $p < 0.01$; ***:

$p < 0.001$. **k** WSP-1 staining of tumour tissue after different treatments. Images are representative of three biologically independent mice. Scale bar: 200 μm. **l, m** Flow cytometry analysis and quantification of the percentage of F4/80⁺ cells in tumours, spleen, and lymph nodes. Data are presented as the means±S.D. ($n = 3$ independent mouse). Statistical differences were calculated using two-tailed Student's t test, *: $p < 0.05$; **: $p < 0.01$; ***: $p < 0.001$. **n** Flow cytometry analysis of M1 (CD 86⁺) and M2 (CD206⁺) cell in tumours, spleen, and lymph nodes, and NK cells (CD49b) in tumours. **o, p** Statistics of tumour number after 15 d of treatment, and bright-field image of tumour after different treatment. **q** Survival curve analysis within 60 d. **r** Immunofluorescence staining of colorectal TUNEL, CD49, GPX4, CD86, and CD206 after 2 d of treatment. Images are representative of three biologically independent mice. Scale bar: 1 mm, 200 μm, 1 mm, 200 μm, 1 mm, and 100 μm from top to bottom.

remained from the tumour site, and this result was also confirmed by the dissection of the organs. According to the intestinal anatomical results, the jejunum and ileum in the lung metastasis model of CRC retained a significant number of probes at 8 h, which was basically consistent with the CRC model in situ (Fig. 6c). Consequently, PfCC had a good effect on targeting metastatic tumours, creating a promising basis for the treatment of tumour metastasis.

### Anti-tumour targeting for orthotopic induced CRC tumours in vivo

Subsequently, by monitoring the changes of pro-inflammatory factors/chemokines in orthotopic CRC mice, the therapeutic efficacy of PfCC to alleviate inflammation was evaluated under the modulation of the photothermal effect combined with endogenous H₂S. FVB mice with in situ CRC were divided into five groups (8 mice in each group), and additionally treated with PBS, PBS + NIR, MCC, MCC + NIR, PfCC+NIR. All the above probes were injected into mice through the tail vein, the concentration was 80 mg/kg, and measured by Co²⁺ concentration. Six h after injection of the probes, the lower abdomen of the mice was treated with NIR (Fig. 7a), three mice were removed from each group, and then blood was subsequently collected for ELISA analysis of inflammatory factors. The results demonstrated that the expression levels of IL-6 and IL-1β were high in the PBS and NIR groups (Fig. 7b), the MCC group was slightly lower, the MCC + NIR group was second, and the PfCC+NIR was the lowest, nevertheless, the change of TNF-α was reversed. This was because the consumption of high-level H₂S in the intestine by Co²⁺ alleviated the intestinal inflammatory environment, and led to a reduction in inflammatory factors. The generated CoS precipitate had a strong photothermal effect, which further killed the tumour and promoted the increase of TNF-α. Furthermore, peptides might regulate the balance of intestinal microorganisms, especially the growth of Desulfovibrio, and further improve the inflammatory microenvironment of intestinal tumours. From the HE results of the intestine, it revealed that the occurrence of intestinal tumours changed the structure of the intestinal crypt (Fig. 7c). As the tumour was formed, some intestinal crypts showed distortion, and the crypt lumens displayed concentrated or scattered tumour cells. The immunostaining results of CD86 and CD206 showed that, the CRC microenvironment was mainly composed of M2 macrophages in PBS group, and M1 macrophages in the MCC group gradually increased after the probe was modified with peptides, M1 macrophages were obviously dominant. This further suggested that the regulation of Co²⁺ on the inflammatory tumour microenvironment could significantly change the phenotype of tumour-associated macrophages, namely, the relative abundance of M1 macrophages increased, and M2 macrophages decreased, while the regulation of intestinal microbiota by peptides further improved the inflammatory tumour microenvironment. After 27 d of treatment, the weight change of the mice was quantified (Fig. 7d). It was observed that the treatment did not result in a significant loss of body weight, thereby further substantiating the in vitro safety of the probe.

As one of the most common malignant tumours, the development of in situ CRC is closely related to chronic intestinal inflammation. Patients with CRC frequently often suffer from ulcerative colitis in the late stages of the disease. Here, the in situ CRC mouse model was constructed by the combined effect of AOM-induced mouse gene mutation and DSS-induced inflammation. When observing in situ CRC tumours, endoscopy is commonly used for qualitative analysis of size or quantitative analysis of tumours after dissection of model mice. To perform quantitative analysis of the tumour in situ in vivo, AI-based deep learning was employed to segment the original tumour map with using the large model SAM, thereby obtaining the segmented tumour map (Fig. 7e–g and Supplementary Fig. 26), creating an algorithm for extracting the pixel coordinates of the tumour region, and fitting a linear equation between the pixel and the actual measured distance of the intestinal tumour (Fig. 7h), finally achieving the quantitative analysis of intestinal tumours. This method could help the following treatment evaluation.

As mentioned above, H₂S plays a crucial role in the development and progression of CRC. So we conducted a quantitative experiment to detect H₂S in tumour by using the WSP-1 probe, which has high selectivity and specificity for H₂S. After reacting with H₂S, the fluorescence intensity of WSP-1 will be enhanced. The main method involved preparing a standard curve, followed by grinding and digesting tumour tissue, adding the WSP-1 probes, measuring the fluorescence signal after a specific incubation period, and calculating the H₂S concentration via the standard curve. The results showed that when 100 μg/mL of H₂S reacted with different concentrations of WSP-1 for 5 min, 50 μg/mL of WSP-1 was sufficient to react with most of the H₂S (Supplementary Fig. 27a). So we chose 50 μg/mL of WSP-1 as the detection concentration of H₂S, and drew a standard curve for the reaction of different H₂S concentrations (Supplementary Fig. 27b and c). It was found that after 5 min treatment with WSP-1, the concentrations of H₂S in tumour tissues treated with PBS, NIR, MCC, MCC + NIR, and PfCC+NIR were about 2.8, 2.5, 1.27, 0.98, and 0.676 mM, respectively(Supplementary Fig. 27d and Fig. 7i). MCC could reduce the H₂S in the tumour by 54.6%, which was caused by the reaction of Co²⁺ with H₂S. The PfCC+NIR group could reduce H₂S by 75.8%, which was significantly different from the MCC + NIR group (Fig. 7j). This may be because the polypeptide affected the generation pathway of H₂S in the tumour. The H₂S staining in the tumour with WSP-1 also qualitatively confirmed this result, and after reducing H₂S, the green fluorescence area in the tumour tissue was significantly reduced (Fig. 7k).

For the possible changes in the tumour microenvironment caused by different treatments, we conducted a series of qualitative and quantitative studies on macrophages and NK cells in tumours, spleen, and lymph nodes (LN). we found that compared with the PBS group, the distribution of F4/80⁺ macrophages of the tumour and lymph in the MCC group gradually increased; compared with MCC + NIR group, the macrophages of the tumour and lymph in the PfCC+NIR treatment group were also significantly increased, indicating that MCC and antimicrobial peptides could induce

macrophages response in tumours (Fig. 7l, m). Since low concentrations of $H_2S$ can promote the polarization of M2 macrophages to M1 macrophages by inhibiting the expression of NF-κβ, we verified the expression of NF-κβ in CT26 cells with different treatments. The flow cytometry results showed that compared with the PBS group, the expression of NF-κβ in the MCC group was slightly reduced, this was because the release of $Co^{2+}$ can initially consume the high concentration of $H_2S$ in the tumour, and after PfCC+NIR treatment, the expression of NF-κβ was the lowest (Supplementary Fig. 28), maybe the intratumoural release of antimicrobial peptides affected the $H_2S$ bacteria production pathway in the tumour, this was further verified by the above-mentioned quantitative detection of $H_2S$ (Fig. 7i). Secondly, we performed analysis of different macrophage types and NK cells within the tumour. The results showed that after MCC treatment, the proportion of M1 macrophages in the tumour increased compared with the PBS group, and after treatment with PfCC+NIR group, the proportion of M1 macrophages in the tumour further increased, while M2 macrophages decreased, this was because low concentrations of $H_2S$ can promote the polarization of M2 to M1 macrophages. And the change in the lymph nodes was just the opposite, which further illustrated that the PfCC may further enhance the innate immune response in the tumour by promoting the migration of M1 macrophages in the lymph nodes into the tumour (Fig. 7n and Supplementary Fig. 29a). Finally, we conducted the analysis of NK cells within the tumour, spleen, and lymph nodes. The results showed that the amount of NK cells in the tumour increased with the treatment of MCC, and further increased after further treatment with the released peptides in PfCC, and appeared decreasing trend in the spleen and lymph (Fig. 7n and Supplementary Fig. 29b). This also demonstrated that PfCC could affect the distribution of NK cells in the tumour and had the preliminary effect of activating the innate immune response in the tumour. Next, the CRC tumours were photographed endoscopically at 6 and 15 d after probe injection. Based on the results of the above algorithmic analysis, it was found that on the 6th day, compared with the PBS and NIR group, the volume and number of tumours in the MCC group were relatively reduced (Supplementary Fig. 30a and b), followed from the MCC + NIR group, and the smallest in the PfCC+NIR group. On the 15th d, this trend was more obvious, and the statistical analysis of the tumour numbers in each group also verified the above result (Fig. 7o). The endoscopic results demonstrated that the colorectal tumours in the PBS group continued to develop and worsen over time, and did not develop significantly in the PfCC+NIR group on the 15th d compared with the 6th d (Fig. 7p). After 60 d of treatment observation, survival curve analysis was performed. It was revealed that (Fig. 7q) mice in the PBS and NIR groups died within 30 d, MCC and MCC + NIR mice died completely around 35 and 52 d, respectively, and only one mouse died in the PfCC+NIR group on the 60th d, which further substantiated that PfCC had a remarkable synergistic therapeutic effect. Two days after the end of treatment, the colons of mice were harvested, and subjected to staining of swiss roll sections (Fig. 7r). The colorectal tumours in the MCC, MCC + NIR and PfCC +NIR groups all exhibited varying degrees of apoptosis and necrosis. Furthermore, the amount of apoptotic and necrotic cells in the PfCC +NIR group was significantly higher than that in the former two groups, which was consistent with the previous data, and indicated that endogenous gas modulation was enhanced in combination with precipitation photothermal effect. GPX4 in the tumour represents the prognosis of the tumour. Compared with the PBS, MCC, and MCC + NIR groups, the pink fluorescence signal of GPX4 was weaker in the PfCC+NIR group, which further showed the superior prognosis of the PfCC+NIR group. The results of macrophage and NK cell immunostaining also showed that after 2 d of treatment, the abundance of M2 macrophages labelled with green fluorescence in the PBS treatment group was relatively large; with the consumption of

endogenous $H_2S$ and the alleviation of the inflammatory tumour microenvironment, the abundance of M1 macrophages labelled with pink fluorescence increased after treatment of MCC, and the change of immunosuppressive microenvironment led to the gradual increase of red fluorescent labelled NK cells in tumours. After treatment of MCC + NIR, the above change trend was more obvious, and with the conjugation of peptide, the abundance of activated NK cells and M1 macrophages increased most significantly. This might be attributed to the antimicrobial peptide's effect on the recruitment and activation of immune cells in the tumour microenvironment, and the gradual recovery of inherent immunity. So PfCC exhibited a notable inhibitory effect on tumour development and progression.

## Modulation of microorganisms at CRC tumour sites based on antimicrobial peptide

Maintaining the homoeostasis of the intestinal microbiota, inhibiting the formation of intestinal inflammatory microenvironment, and remodelling host immune function are key factors in inhibiting the occurrence and development of inflammatory bowel disease and CRC[46–48]. To investigate the antibacterial effect of antimicrobial peptide on bacteria (taking *Escherichia* and *Vibrio* as examples), different concentrations of antimicrobial peptide were mixed with the bacterial cultures, and incubated for 1 h. The colonies on the LB agar plates were recorded. The results showed that as the concentration of antimicrobial peptide increased, the growth of *Escherichia* and *Vibrio* was significantly inhibited. Moreover, even at a concentration as low as 10 μg/mL, it could still have a good antibacterial effect (Supplementary Fig. 31). Subsequently, we further explored the effects of different probes on different bacteria. The results showed that both antimicrobial peptide and PfCC had significant inhibitory effects on Escherichia and Vibrio, and were slightly better than penicillin (Fig. 8a, b). This further demonstrated the great potential of antimicrobial peptides in regulating bacterial proliferation in vivo. So the regulatory effect on the intestinal microenvironment, and the treatment effect of CRC tumour in vivo were investigated. The faecal microbiota of mice receiving different treatments was analysed at different stages. According to 16S rRNA identification, among the samples of different treatment groups, the α-diversity of the community did not change significantly (Fig. 8c). Based on β-diversity non-metric multidimensional scaling (NMDS) analysis, CT26 tumour mice treated with PBS, MCC, PfCC showed obvious changes in community clustering (Fig. 8d). At the class level, the two main types of bacteria including bacilli and bacteroidia were observed in the PfCC group (Supplementary Fig. 32). At the genus level, compared with the MCC group, the abundance of *Desulfovibrio* and *Escherichia_Shigella* decreased in the PfCC groups at the end of the experiment (Fig. 8e-g). Additionally, the beneficial *Akkermansia* and *Bifidobacterium* may show an increasing trend (Supplementary Fig. 33). At the family level, the abundance of *Desulfovibrionaceae* was found to be significantly lower in the MCC and PfCC group, whereas the abundance of *Bifidobacteriaceae*, *Ruminococccaceae*, *Akkermansiaceae*, and *Lachnospiraceae* was significantly higher (Fig. 8h). This might be attributed to the fact that $Co^{2+}$ consumed the high $H_2S$ concentration in CRC tumour sites, weakened the inflammatory environment, and inhibited the growth of inflammation-related flora to a certain extent. Furthermore, antimicrobial peptides had a significant inhibitory effect on the gram-negative bacteria *Desulfovibrio* and *Escherichia*, which in turn promoted the growth and proliferation of the beneficial bacteria. This further demonstrated the regulatory effect on intestinal microflora-tumour flora, which achieved synergistic $Co^{2+}$ to regulate intestinal microflora to improve the inflammatory immunosuppressive tumour microenvironment, thus combining with precipitation photothermal therapy for the treatment of CRC in situ.

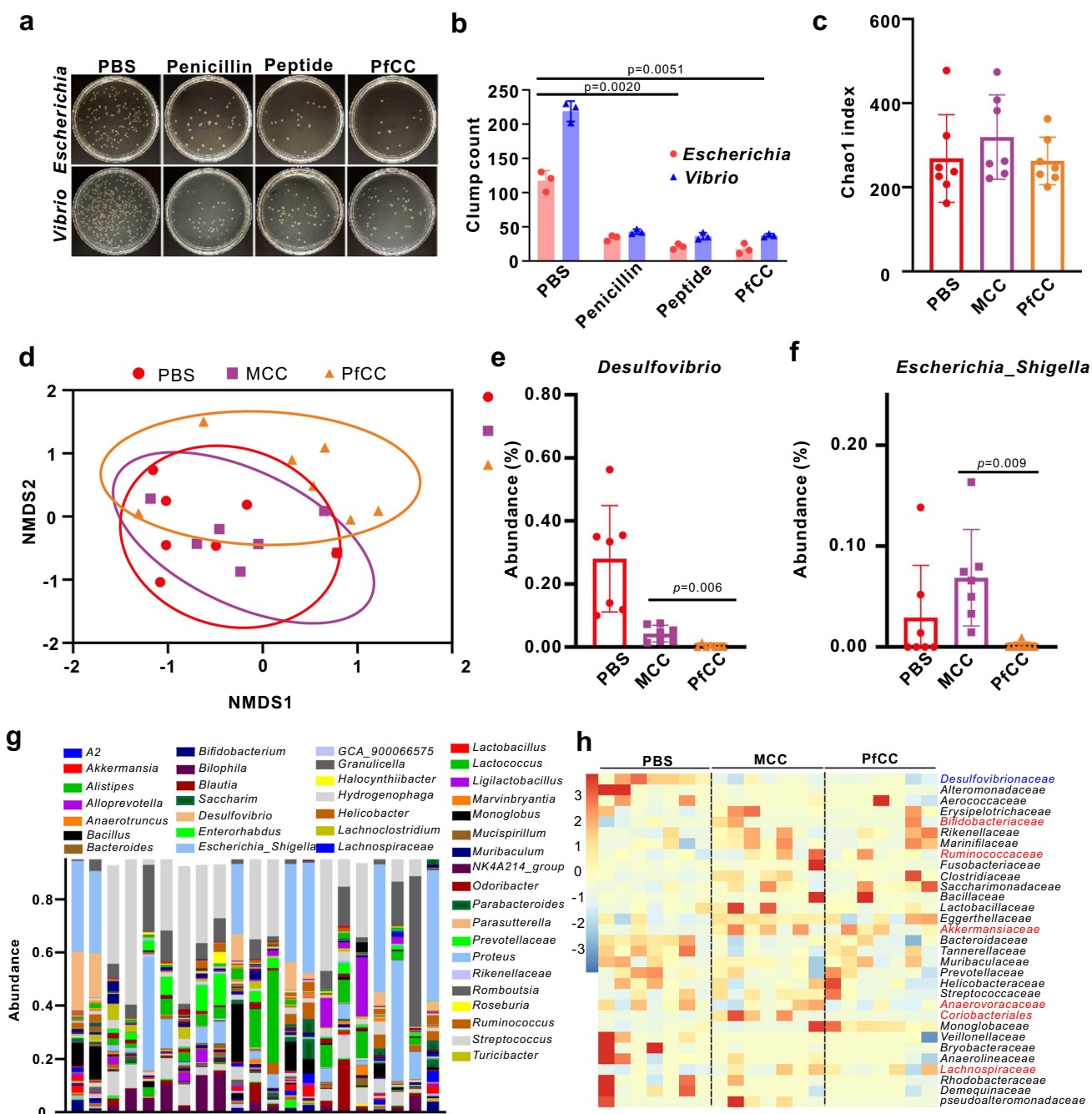

**Fig. 8 | 16S rRNA sequencing of intestinal flora regulation by different probes. a, b** The inhibitory effects of different probes on the growth of *Escherichia coli* and *Vibrio* bacteria. Data are expressed as mean ± SD (*n* = 3 independent samples). Statistical differences were calculated using two-tailed Student's t test, *: *p* < 0.05; **: *p* < 0.01; ***: *p* < 0.001. **c** Microbial α-diversity expressed in Chao1 index. Data are expressed as mean ± SD (*n* = 7 independent samples). **d** NMDS analysis of microbial β-diversity based on Bray-Curtis distance. NMDS: Non-metric Multidimensional Scaling. **e, f** Changes in the relative abundance of *Desulfovibrio* and *Escherichia_Shigella* bacteria during 48 h of different probe treatments. Data are expressed as mean ± SD (*n* = 7 independent samples). **g** Histogram of the relative abundance of gut microbiota at the level of genus. **h** Heat map of the relative abundance of gut microbiota at the family level.

## Discussion

This work successfully synthesized a biomimetic nano-platform (PfCC) leveraging peptide modified-CRC cell membrane coating and a degradable cobalt-based MOF for synergistic in situ CRC therapy. PfCC achieved tumour-targeted enrichment through homologous homing effect, where the acidic degradation triggered three coordinated anti-tumour mechanisms: 1). $H_2S$ scavenging: released $Co^{2+}$ consumed endogenous $H_2S$ via precipitation (CoS). 2). Downregulation of $H_2S$ production: the released peptides modulated the tumour microbiome

such as *Desulfovibrio*, sustaining low $H_2S$ levels. 3). Immunomodulation & PTT: reduced $H_2S$ inhibited pro-tumour transcription factors, promoting M2-to-M1 macrophage polarization. Concurrently, CoS precipitate formation enabled PTT, further reversing immunosuppression and activating innate immunity.

While PfCC demonstrated compelling efficacy in preclinical models, its path to clinical application faces key challenges. Reproducible large-scale production of cell-membrane vesicles with consistent peptide insertion efficiency and coating homogeneity requires

robust GMP-compliant processes. Additionally, standardizing extrusion methods and ensuring batch-to-batch consistency are critical hurdles. For the long-term safety, as the above study, the 7 d metabolic rate of cobalt ions in the body basically reached 90%, indicating that there were no potential long-term toxicity.

Here, PfCC offered distinct advantages over existing $H_2S$-modulating or photothermal strategies. Compared with conventional $H_2S$ Inhibitors such as small-molecule $H_2S$ inhibitors (e.g., AOAA, PAG), they often suffer from poor tumour specificity, systemic toxicity, and inability to address both $H_2S$ overproduction and excess existing $H_2S$. But PfCC uniquely combined simultaneous $H_2S$ scavenging and sustained production blockade (peptide/microbiome modulation) within the tumour microenvironment (TME), enhancing efficacy and potentially reducing off-target effects. For traditional PTT Agents, while inorganic PTT agents such as Au nanorods and CuS nanoparticles are potent, their non-degradability raises long-term safety concerns. Whereas CoS precipitate provided localized PTT only upon TME-specific degradation, minimizing background toxicity. Furthermore, the PTT effect was not the sole mechanism but synergized with immunomodulation triggered by $H_2S$ depletion.

Beyond the therapeutic platform, this study pioneered the use of AI large model learning to assist in developing convex hull algorithms for the quantitative assessment of in situ CRC tumours. This approach will provide a robust, objective tool for evaluating therapeutic response in vivo and establishes a valuable algorithmic framework adaptable to other in situ tumour models.

# Methods

## Materials
N, N-dimethylformamide, triethylamine and anhydrous methanol were purchased from Sinopharm Chemical Reagent Co., Ltd.; cobalt nitrate hexahydrate ($Co(NO_3)_2 \cdot 6H_2O$) and dimethylimidazole were purchased from Aladdin; hydrogen sulphide standard solution was purchased from Macklin; WSP-1 was purchased from Beijing Solarbio Science & Technology Co., Ltd.; membrane and cytosol protein extraction kit, Dil and lyso-tracker green and annexin V, FITC apoptosis detection kit were purchased from Shanghai Beyotime Biotechnology Co., Ltd.; D-luciferin, potassium salt was purchased from Yeasen Biotechnology (Shanghai) Co., Ltd.; cell counting kit-8(CCK-8) was purchased from Nanjing Novazan Biotechnology Co., Ltd.; pierce BCA protein assay kit was purchased from Thermo Fisher Scientific; bouin fixative was purchased from Goldclon (Beijing) Biotechnology Co., Ltd.; FITC anti-mouse CD86 Antibody, APC anti-mouse F4/80 Antibody, and PE anti-mouse CD206 were purchased from BioLegend; PE Rat anti-Mouse CD49b Antibody was purchased from Absin; CT26 cell was purchased from China Centre for Type Culture Collection (Wuhan, China); Luc-CT26 cell was purchased from Fenghui Biological Technology Co., Ltd.; *Escherichia coli* was kindly provided by Dr. Xiao-Ling Lei; the 7-week-old FVB female mice (SPF grade), 4-week-old BALB/c female mice (SPF grade) and 5-week-old KM mice were purchased from Beijing Vital River Laboratory Animal Technology Co., Ltd. All mice were housed in a stable environmental conditions (room temperature, $22 \pm 1\,°C$, relative humidity, 40-70% and a 12 h light-dark cycle), and all mice had access to food and water ad libitum. All animal experiments were approved by the Animal Experimentation Ethics Committee of Huazhong University (IACUC Number: 4208).

## Instrumentation
The fluorescence imaging system was constructed by our laboratory; MDL-III-808-2.5 W laser (Changchun New Industries Optoelectronics Tech. Co., Ltd, China) was used for laser irradiation; electron microscopy characterization was performed on Hitachi 120 kV HT7700 transmission electron microscope (Hitachi, Japan); particle size and zeta potential of the probe were characterized by Nano-ZS90 nanometre (Malvern, UK); vacuum concentrator (Ependorf, Germany) was

to purify the probe; absorption spectra of probe was proceeded by UV-2550 ultraviolet-visible spectrophotometer (Shimadzu, Japan); x'pert3 power X-ray diffractometer (Panalytical BV, Netherlands) was used for the structure analysis of nanoparticle; WFX-200 atomic absorption spectrophotometer (Beijing Beifen-Ruili Analytical Intrument Co., Ltd., China) was to detect concentration of Co; quantitative analysis of probe-induced apoptosis were obtained through Cyto FLEX flow cytometer (Beckman Coulter, USA); FV3000 confocal microscope (Olympus, Japan) was to acquire cell fluorescence staining imaging; changes of characteristic functional groups during synthesis were verified by Vertex 70 infra-red spectrometer (Bruker, Germany); absorbance detection of well plate was carried out by flexStation3 microplate reader (Molecular Devices, USA).

## Synthesis of C
The synthesis of C involved the preparation of two different liquids: liquid A, which contained 12 mg of $Co(NO_3)_2 \cdot 6H_2O$ dissolved in 20 mL of anhydrous methanol, and liquid B, which consisted of 30 mg of dimethylimidazole and 5 μL of triethylamine dissolved in 5 mL of anhydrous methanol. Subsequently, liquid A was added to liquid B with thoroughly stirring. After reacting for 4 h at room temperature, the system was centrifuged at 15422 g for 15 min, washed several times with anhydrous methanol, collected, and precipitated for later use.

## Synthesis of antimicrobial peptide and phospholipid coupling
The antimicrobial peptide $(IRVK)_3$-$NH_2$ was synthesised by solid-phase synthesis of Fmoc peptides. The Rink amide MBHA resin was immersed in DMF for 3 h to activate the swelling, and five-fold equivalent of the amino acid was activated by HBTU/HOBT for 45 min for the cross-linking reaction. After the reaction, the Fmoc protecting group was removed by piperidine/DMF (20%, v/v), and the aforementioned steps were repeated until $(IRVK)_3$-$NH_2$ was obtained. After 1 mg of DPPC-$PEG_{2000}$-COOH was activated by NHS for 0.5 h, 1.2-fold equivalent $(IRVK)_3$-$NH_2$ was added for 2 h and then centrifuged over a desalting column to obtain purified antimicrobial peptide-coupled phospholipid DPPC-$PEG_{2000}$-$(IRVK)_3$.

## Cobalt-based MOF functioned by antimicrobial peptide-modified cell membrane (PfCC)
~$1.0 \times 10^8$ CT26 CRC cells were washed with PBS, treated with trypsin digestion, and centrifuged at $4\,°C$ (250 g, 5 min). After removing the supernatant, 2 mL of membrane protein extraction reagent A containing PMSF (20 μL) was added to cell precipitation. Once fully suspended, cells were placed in an ice bath for 10 min, frozen and thawed 3 times in liquid nitrogen (2 min) and $37\,°C$ (5 min). After centrifugation (700 g, 10 min) at $4\,°C$, the nucleus and intact cells were removed to collect the precipitate, and they were dissolved in PBS for continuous centrifugation (14000 g, 30 min) at $4\,°C$. The cell membrane fragments were precipitated, and the membrane protein concentration was detected by BCA protein assay kit, then collected and lyophilized into powder for later use.

One millilitre of C (4 mg/mL) and 1 mL of CRC cell membrane (1 mg/mL) were mixed and stirred in an ice bath for 1 h, then repeatedly passed through an Avanti microextruder (400 nm polycarbonate porous membrane), and centrifuged to remove excess cell membrane fragments, that is, to obtain membrane-coated C(MCC). One point five millilitre of the above MCC solution was added into 0.2 mL of DPPC-$PEG_{2000}$-$(IRVK)_3$ (1 mg/mL) solution, incubated (2 h) and centrifuged (8000 r, 10 min) to obtain the PfCC. The same method could be employed to obtain the fluorescent dye Cy5.5-labelled PfCC. Then, the content of Co in PfCC were quantified. First, 1 mL of PfCC were dried by freeze-drying, and weighed to obtain 4.75 mg of solid powder. With using an atomic absorption spectrophotometer, the standard curve for cobalt was plotted, and the cobalt ions were quantified in PfCC

## Stability and haemocompatibility of PfCC

The lyophilised PfCC was dissolved in PBS, DMEM, and DMEM + 50% serum to measure the particle size, PDI, and zeta potential at room temperature, and the detection was repeated every 1 d to investigate the stability of the probe.

One point five millilitre of fresh blood from 4 six-week-old KM mice were collected into EDTA K2 anticoagulation tubes. After removing the supernatant by centrifugation (700 g, 5 min), the precipitate was washed several times with PBS. Ten microliters of suspension was added to 250 μL of PBS containing MCC (0.05, 0.2, 0.4, and 0.8 mg/mL). Here, PBS and ultrapure water were used as negative and positive controls, respectively. The above blood samples were incubated in a 5% $CO_2$ incubator (37 °C, 8 h), centrifuged at 2039 g for 5 min, and the supernatant was collected to measure the absorbance value at 577 nm. Finally, the cell pellet was processed into cell smears, stained with giemsa, and observed the morphology under microscope. So the haemolysis rate of red blood cells was calculated as follow (2).

$$\text{Hemolysis rate (\%)} = (\text{OD}_{\text{sample}} - \text{OD}_{\text{PBS}})/(\text{OD}_{\text{water}} - \text{OD}_{\text{PBS}}) \times 100\%$$
(2)

## Photothermal effect of H₂S-induced precipitate from probes

Four aliquots of 1 mg/mL MCC were placed in PBS at pH 6.5 for 0.1, 0.5, 1, and 2 h. The reaction solution was diluted and divided into equal parts, and different concentrations of $H_2S$ standard solution (1.5, 1.6, 2.0, 2.1 mM) were added for 2 h, and centrifuged (94 g, 5 min) to obtain the precipitate, which were irradiated with an 808 nm laser for 5 min, and the temperature change of the samples were recorded by infra-red thermal imaging camera.

## Cytotoxicity assay of MCC and AMP

~$1 \times 10^5$ CRC CT26 and 4T1 cells in logarithmic growth phase were inoculated into 96-well plates and cultured overnight in a 5% $CO_2$ incubator (37 °C). After removing the culture medium, 200 μL of serum-free medium containing different concentrations of MCC (0, 0.006, 0.012, 0.025, 0.05, 0.1, 0.2, 0.4, and 0.8 mg/mL) and AMP (0, 0.01, 0.02, 0.04, 0.08, 0.16, 0.32, 0.64, and 1.28 mg/mL) was added, and the probe concentration was determined by that of C. Five parallel wells were set up in each group. After 24 h, the culture medium was aspirated and washed 3 times with PBS, another 20 μL of CCK-8 solution was added to fresh serum-free medium, incubated for 4 h and the absorbance was measured at 450 nm by a microplate reader. The cytotoxicity of AMP was added for another 48-h group. Cell viability was calculated according to the following formula:

$$\text{Cell viability (\%)} = (\text{OD}_{\text{experimental group}} - \text{OD}_{\text{blank group}})/$$
$$(\text{OD}_{\text{control group}} - \text{OD}_{\text{blank group}}) \times 100\%$$
(3)

About $1 \times 10^5$ CRC CT26 cells in the logarithmic growth phase were inoculated into 96-well plates and incubated overnight at 37 °C in a 5% $CO_2$ incubator. After the culture medium was removed, the groups were set as PBS, NIR, MCC, MCC + NIR, PfCC, PfCC+NIR, and 200 μL of serum-free medium with different probes was added. The probe concentration was quantified by that of C, and all of them were 0.1 mg/mL. Five parallel wells were set up in each group. After 24 h, the culture medium was aspirated and washed 3 times with PBS, fresh serum-free medium was added with another 20 μL of CCK-8 solution. Absorbance at 450 nm was measured with a microplate reader, and the cell viability was calculated after 4 h of incubation.

After culturing CT26 cells under the same conditions, 200 μL of serum-free medium containing PfCC (100 μg/mL, probe concentration was quantified by that of C) was added to a 96-well plate. 4 h later, the culture medium was removed, washed 3 times with PBS, and added

with 10 μL of $H_2S$ (20, 40, 80 μg/mL) standard solution. After 2 h incubation, the medium was removed and washed 3 times by PBS. Laser irradiation (808 nm, 0.75 W/cm²) was performed for 5 min while adding the fresh medium. For another 24 h incubation, 20 μL of CCK-8 solution was added to each well. Absorbance at 450 nm was measured by a microplate reader and cell viability was calculated after 4 h of incubation.

## Explore the effects on intracellular H₂S and cell-targeted colocalization and phagocytosis

About $5 \times 10^5$ CT26 cells were inoculated into confocal dishes and cultured in a 5% $CO_2$ incubator (37 °C, 12 h). After removing the culture medium and setting up the control, $H_2S$ (20 μg/mL), PfCC+H₂S group, the cells were incubated for 4 h, rinsed thoroughly with PBS, and observed under a confocal microscope.

~$5 \times 10^5$ CT26 cells were seeded into confocal dishes and incubated in a 5% $CO_2$ incubator (37 °C, 12 h). After aspirating the culture medium, the cells were cultured for different time (0.5, 1, 2, 4, 6, 8, and 10 h) and concentrations of Dil-labelled PfCC (20, 50, 100 μg/mL; Dil: 1 μg/mL). Then Lyso-Tracker green and hoechst 33342 were added, washed thoroughly with PBS, and observed under a confocal microscope.

## Cell therapy with probes

About $5 \times 10^5$ CT26 cells were seeded in confocal dishes, incubated in a 5% $CO_2$ incubator (37 °C, 12 h), and removed the culture medium. One milliliter of serum-free culture medium containing MCC and PfCC was added. After removing the culture medium at 5 h, the cells were washed 3 times, added with PBS, and treated with the laser (1.0 W/cm²). Subsequently, calcein and PI were successively added for 5 min. After the staining solution was removed, the cells were washed 3 times with PBS, and observed under a confocal microscope. After incubation with PfCC, the cells treated with different concentrations of $H_2S$, and added with Annexin V-FITC/PI apoptosis staining solution. 15 min later, the cells were washed 3 times with PBS, and apoptosis analysis was performed.

## Inhibition of migration and invasion of CT26 cells

~$1 \times 10^5$ CT26 cells were inoculated in 0.1 mL of serum-free 1640 culture medium in the upper chamber of Transwell, with a polycarbonate membrane pore size of 8 μm, and 0.5 mL of serum-containing 1640 culture medium in the lower chamber. PBS, MCC, and PfCC (the above probe concentration was calculated as cobalt-based MOF and was 0.1 mg/mL, 0.1 mL) were respectively added in the upper chamber. After incubation of 4 h, NIR treatment was performed, and incubated for another 24 h. Subsequently, the ester film of the upper chamber was removed from the transwell, and the cells in the chamber that did not penetrate it were removed by wiping with a cotton swab, while the cells in the small chamber were fixed in anhydrous methanol for 10 min. The obtained cells were washed 2 times with PBS, air-dried, stained with 0.1% crystal violet for 10 min, and continued to wash 2 times with PBS. Cells entering the upper chamber were counted. In addition to the aforementioned description, the invasion assay requires Matrigel coating of the upper chamber.

## Safety of PfCC in vivo

Twenty four-week-old male BALB/c mice (SPF grade) were randomly divided into four groups, two of these groups were killed 45th d after tail vein injection of 200 μL of PfCC and PBS, while the third and fourth groups were killed in 1st and 20th d after injection of 200 μL of PfCC, respectively. Fresh blood was collected from each mouse for routine blood and biochemical analyses, and the corresponding organs were weighed to calculate organ index and analysed by HE staining. Additionally, the changes in body weight of the PBS and 45 d probe-treated

groups were monitored and recorded every two days.

$$\text{Organ index} = \text{organ weight}/\text{mouse body weight} \times 100\% \quad (4)$$

### Targeted fluorescence imaging in CRC mice

Subcutaneous tumours: Ten five-week-old male BALB/c mice (SPF grade) were randomly divided into two groups, five of them were inoculated with about $1 \times 10^6$ CT26 cells, and the others were inoculated with about $1 \times 10^6$ CT26 and 4T1 cells on both sides. When the tumour volume increased to 50–120 mm³, mice were injected with 200 μL of Cy5.5-labelled PfCC (5 mg/mL) through the tail vein for continuous fluorescence imaging at different time (0, 2, 4, 6, 12, 24, 60, and 72 h).

Fifteen five-week-old male BALB/c mice (SPF grade) were randomly divided into 5 groups and inoculated with $1 \times 10^6$ Luc-CT26 cells. When the tumour volume grew to 50–120 mm³, 200 μL of Cy5.5-labelled PfCC (5 mg/mL) was injected into the mouse through the tail vein and the fluorescence imaging was performed at different times (0.1, 6, 12, 60, and 72 h). Additionally, the fluorescein potassium salt (15 mg/kg) should be injected 15 min in advance to allow bioluminescence imaging of the tumour site. After dissection, blood was collected for blood half-life analysis, and the organs, tumours, and three tumour slices cutting into parts from outside to inside were performed for fluorescence and bioluminescence imaging. The fluorescence intensity of the corresponding organs was also counted. During the experiments, the mice were anesthetized and fixed on the imaging platform by the respiratory anaesthesia system. When the tumour volume reached 1000 mm³ or at the end of the experiment, the mice were euthanised in accordance with animal welfare law.

In situ tumours: twenty-four orthotopic CRC model FVB mice were randomly divided into 8 groups, and 200 μL of Cy5.5-labelled PfCC (5 mg/mL) were injected into the body through the tail vein. At different time (0.1, 2, 4, 6, 8, 12, 24, and 36 h), the in vivo, abdomen removed with the skin, intestine, colorectum, and the organs and blood at the corresponding time were took for fluorescence imaging.

Metastatic tumour: to establish a CRC lung metastasis model, 9 BALB/c nude mice were injected with $2 \times 10^6$ Luc-CT26 cells through the tail vein. After 15 d, 200 μL of Cy5.5-labelled PfCC (5 mg/mL) was injected into the body through the tail vein, and fluorescence imaging was performed in vivo and organs after intraperitoneal injection of fluorescein potassium salt (15 mg/kg) in mice at the 4th, 8th, and 24th h. At the end of experiments, the mice were sacrificed and the corresponding lung tissues were fixed by bouins and observed by HE staining.

### In vivo therapy experiments for subcutaneous tumours in mice

BALB/c mice inoculated with CT26 cells were treated with (I) PBS, (II) PBS + NIR, (III) MCC, (IV) MCC + NIR, (V) PfCC+NIR (the above probe injections were all injected into the tail vein), and the temperature changes at the tumour site were recorded by thermal imaging. After the corresponding treatment, the mice were sacrificed, and kept at constant temperature for 2 h. The tumours were removed, washed several times, and immersed with 4% paraformaldehyde, fixed for 48 h, and then embedded to make sections. Finally, HE, TUNEL, CD86, CD206, and CD49 immunofluorescence staining were performed, and the results were observed under the microscope.

twenty-five BALB/c mice inoculated with CT26 cells were divided into 5 groups and treated with (I) PBS, (II) PBS + NIR, (III) MCC, (IV) MCC + NIR, (V) PfCC+NIR (each time point was consistent with the above experiment). Additionally, PBS, MCC (60 mg/kg) and PfCC (60 mg/kg) were injected through the tail vein, and the injection volume was 200 μL. Weight and tumour volume changes were recorded every 2 d by digital vernier calipers and weight scales. After 28 d (or when the tumour volume reached 1000 mm³), the mice were

sacrificed, the tumours were weighed, and the tumour index was calculated as,

$$\text{Tumour index} = \text{tumour weight}/\text{mouse body weight} \times 100\% \quad (5)$$

The same treatment method was added to a PBS and AMP group for 20 d of experimental observation.

### Construct the orthotopic CRC mouse models

A CRC model was constructed by using azomethane (AOM) in combination with dextran sodium sulphate (DSS). FVB/N mice (female, 6–8 weeks old, >18 g) were randomly divided into two groups. The mice in group I were injected intraperitoneally with AOM (10 mg/kg), after 7 d, the mice were offered to drink DSS (2.5%, 5 mL/piece/d) solution for 7 d, and purified water for another 14 d, which corresponded to one cycle, and the mice were induced with DSS inflammation had to undergo two cycles. Mice in group II were intraperitoneally injected with the same volume of saline as AOM, and continuously drank purified water. All mice were maintained in a barrier environment, and fed freely except for the DSS induction period. The state (whether there was blood in the stool, malaise, etc.) and weight of the mice were recorded for 8 weeks. At the end of the experiment, the induction efficiency of CRC was evaluated by endoscopic observation of colorectal tumour formation, and tumour pictures from different angles were taken to construct algorithm and evaluate the volume of the tumour in situ.

### In vivo treatment of CRC in situ

Tumour size was assessed endoscopically based on the tumour size conversion algorithm established in the previous phase. When the tumour volume reached 100 mm³, and 1–3 tumours appeared in the colorectum, the mice with in situ CRC were randomly divided into 5 groups, each group containing 10 mice. The groups were as follows: (I) PBS, (II) PBS + NIR, (III) MCC, (IV) MCC + NIR, and (V) PfCC+NIR. The above probes were injected through the tail vein, and 12 h later, 3 mice in each group were anesthetized with isoflurane, blood was collected from the orbits, and the concentrations of inflammatory factors in different groups were detected by ELISA method. In addition, HE, CD86 and CD206 immunofluorescence staining were performed. Subsequently, the state of the mice was observed, and the weight of the mice was recorded (every other day). On the 2nd day, one mouse colorectum in each group was subjected to WSP-1, TUNEL, CD49, CD86, and CD206 immunofluorescence staining. On the 6th and 27th d, the colorectum of the mouse was observed endoscopically to evaluate the effect of tumour treatment (the number of colorectal tumour globules and tumour size). On the 27th day, one mouse was dissected and the other mice were continued to be fed for observation.

### In vivo detection of H₂S

Drawing of the standard curve: different concentration $H_2S$ (0, 10, 20, 40, and 100 μg/mL) were reacted with (50 μg/mL) WSP-1 probes. After treatment with PBS, NIR, MCC, MCC + NIR, and PfCC+NIR, The mice with in situ CRC tumours were taken for grinding, lysing, incubating with (50 μg/mL) WSP-1 probes for 5 min, then centrifugated (2000 r, 5 min) for fluorescence emission spectrum detection.

### Flow cytometry analysis of macrophages, NK cells, and NF-κβ in vivo after different treatments

After treatment with PBS, MCC, MCC + NIR, and PfCC+NIR, The mice with in situ CRC tumours were taken for grinding, centrifugation (2000 r, 5 min), red blood cell lysis (1 min), and PBS washing (2 times), then they were evenly divided into three parts, one was added with APC-F4/80, FITC-CD68, and PE-CD206; one was added with PE-CD49b; the last was added with NF-κβ p65-FITC, incubated for 20 min, then all were added with DAPI for 5 min, and detected with Cytoflex.

### 16S rRNA analysis of intestinal contents flora in mice after probe treatment

Twenty-one FVB/N mice with orthotopic intestinal tumours were randomly divided into three groups, and after 48 h of PBS, MCC, and PfCC treatment, a certain amount of intestinal contents were quickly collected from the dissected mice, which were quick-removed, frozen with liquid nitrogen and stored at -80 °C for sequencing analysis.

### Construction of AI deep learning

Firstly, we built a deep learning environment based on Python, which was mainly divided into the 5 parts. 1). Installed Pycharm and Anaconda3. 2) Created a virtual environment in Anaconda3 and installed Python: after installing Anaconda, we logged in the Anaconda Prompt, Conda create -n environment name and pthon = 3.10, then continuing running. 3). Installed Pytorch, that was, entering Conda activate environment name in the anaconda prompt, entering the previously created virtual environment and the Pytorch official website command to run. 4). Opened the project with Pycharm, conFig.d the virtual environment, and installed the corresponding Opencv, Matplotlib, Scipy, segment_anything and other dependent packages in our created environment based on the Anaconda Prompt. 5). There were three models (h, l, b) in the demo.py file, among which the h model requires the largest video memory and has the best segmentation effect, and the b model requires the smallest video memory and has the worst segmentation effect. Select the corresponding model to run.

After building the deep learning environment, we started training the data. 38 groups of tumours were photographed endoscopically, and the actual tumour size was determined by dissecting the intestines of mice. Through deep machine learning, the visualization interface was designed based on the open source large-scale model segment anything model (SAM), with the lesion area automatically segmented according to the model prompts. After generating the lesion area mask, the convex hull algorithm was to calculate the convex hull of the tumour region and extract the boundary pixel coordinates, with traversing to calculate the maximum pixel distance of the region. We recorded the pixel distance automatically calculated by the algorithm and the actual measured distance of the anatomical intestine, the pixel distance obtained by the algorithm was used as the $X$-axis and the actual distance as the $Y$-axis, and the linear regression model was fitted to obtain the standard curve.

### Statistics and reproducibility

Experiments in this manuscript were performed with at least three replicates. All values were presented as means ± s.d, exact $p$ values were provided in the figures, and values with $P < 0.05$ are considered significant. In this study, the sample size was determined based on prior experimental experience and standard practices, statistical methods were not used to predetermine the sample size, and we did not exclude data. Detailed descriptions of the experimental methods were provided to ensure that other researchers can replicate the experiment and verify the reproducibility of the results. Statistical analysis was performed with GraphPad Prism (ver. 8.3.0.538) and Origin 2021(ver. 9.80.200). The three-dimensional images were created with Autodesk 3ds Max (ver. 21.0.845). The plots were assembled with Coreldraw x8 (ver. 18.0.0.450). Flow-cytometry data were analysed with FlowJo VX (ver. 10.0.7.2). ImageJ (ver. 1.4.3.67) were used to analyse fluorescent imaging in vivo. Confocal images were analysed with FluoView31S (ver.2.3.1.163). AI deep learning environment was constructed with PyCharm (ver. 252.26199.168). Comparison between two groups was performed using two-tailed Student's t test.

### Reporting summary

Further information on research design is available in the Nature Portfolio Reporting Summary linked to this article.

## Data availability

The authors declare that all data generated or analysed during this study are included in this published article/supplementary Information/Source Data file. The 16S rRNA gene sequencing data generated in this study have been deposited in the Sequence Read Archive under accession number PRJNA1339001. Source data is available for Fig. 2b–d, Fig. 2f–i, Fig. 3b–f, Fig. 4b–d, Fig. 4f, Fig. 5b–d, Fig. 5h–l, Fig. 6f, Fig. 7b, Fig. 7d, Fig. 7h–j, Fig. 7m, n, Fig. 8b–g, and Supplementary Fig. 1–12, Supplementary Fig. 15–18, Supplementary Fig. 21, 22, Supplementary Fig. 27, Supplementary Fig. 29, 30, Supplementary Fig. 32, 33. Source data are provided with this paper.

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

## Acknowledgements

This work was supported by the National Natural Science Foundation of China (Grant No. 62305121 to K. C., 62375093 and 62575109 to Y.-D. Z.), and the China Postdoctoral Science Foundation Funded Project (Grant No. 2023T160247 to K. C.), Technology Innovation Programme of Hubei Province (2024BCB058 to J.H. Z.), the Natural Science Foundation of Hubei Province of China (2025AFB559 to J.-X. F.), China Ageing Development Foundation - Elderly Navigation Programme Research Fund (193 to Y.-D. Z.). We also thank the Analytical and Testing Centre (HUST), the Research Core Facilities for Life Science (HUST) for the help of measurement.

## Author contributions

Conceptualization: K.C. and F.Z. Methodology: K.C., F.Z., G.-P. W., J.X. and X.-L.L. Investigation: K.C., F.Z., X.-L.L., X.T. X., and J.-H.Z. AI analysis: Y.-B.G., G.-P.W. and K.C. Visualization: K.C. and F.Z. Supervision: K.C., B.L., J.-X.F., G.-P.W. and Y.-D.Z. Writing-original draft: K.C., F.Z., and Y.-D.Z. Writing-review & editing: K.C., F.Z., J.-X. F., J.X., and Y.-D.Z.

## Competing interests

The authors declare no competing interests.

## Additional information

[1]Britton Chance Center for Biomedical Photonics at Wuhan National Laboratory for Optoelectronics-Hubei Bioinformatics & Molecular Imaging Key Laboratory, Department of Biomedical Engineering, College of Life Science and Technology, Huazhong University of Science and Technology, Wuhan, Hubei, P. R. China. [2]Department of Chemistry, The Chinese University of Hong Kong, Shatin, Hong Kong SAR, P. R. China. [3]Department of Oncology, Huanggang Central Hospital of Yangtze University, No.126 Qi'an Road, Huanggang, Hubei, P. R. China. [4]Hubei Clinical Medical Research Center of Esophageal and Gastric Malignancy, Huanggang City, Hubei, P.R. China. [5]These authors contributed equally: Kai Cheng, Fang Zhang, Jia-Hua Zou. ✉e-mail: zydi@mail.hust.edu.cn; jiangxia@cuhk.edu.hk; jxfan@hust.edu.cn

