## [Transparent Peer Review file · Nature Communications]

Peptide-Functionalized Membrane Camouflage for Endogenous H₂S-Induced Photothermal Immunotherapy of Orthotopic Colorectal Cancer

Corresponding Author: Professor Yuan-Di Zhao

Version 1:

Reviewer comments:

Reviewer #2

(Remarks to the Author)

This manuscript presents a biomimetic nano-platform by utilizing antimicrobial peptide-functionalized CRC cell membranes to encapsulate a cobalt-based metal-organic framework (PfCC) for the multimodal treatment of colorectal cancer (CRC). This therapeutic strategy combines photothermal therapy and immunomodulation of tumor microenvironment via activating the innate immune response and regulating the endogenous H₂S level and the balance of intestinal flora, providing a new idea for CRC treatment. The authors comprehensively validated the nanoplatform from synthetic characterization, in vitro cells and in vivo animals. In particular, the quantitative analysis of in situ CRC by AI deep learning models (e.g., SAM) provides a new approach for the assessment of tumor treatment effects. However, some problems should be resolved before considering publication.

Major issues:

1. The content of antimicrobial peptide and Co in PfCC needs to be determined.
2. The regulatory effects of antimicrobial peptide and PfCC on different bacteria need to be determined by in vitro assessment.
3. In vitro cellular experiments, including the effect of H₂S on the photothermal effect of PfCC, staining of live and dead cells, and apoptosis assay, all of which need to include the experimental groups of H₂S only and H₂S + PfCC without NIR as controls.
4. In in vitro anti-tumor assay, the quantitative analysis data of macrophages and NK cells should be provided to demonstrate the regulation effect of PfCC on the innate immune response.
5. Details of AI algorithms (e.g., model parameters, training data) need to be supplemented to enhance the reproducibility of results.
6. The discussion section is only a brief conclusion. It is recommended to add the potential challenges of PfCC in clinical translational research (e.g., feasibility of large-scale production, long-term safety in vivo) as well as comparisons with other H₂S-modulated or photothermal therapies, highlighting the significance of this study.

Minor issues:

1. English writing should try to use short sentences to enhance readability. Some sentences of this manuscript are long and it is recommended to simplify them.
2. An n-dash not hyphen should be used to indicate a range of numbers. For example, "2-6" should be correct to "2–6".
3. FTTC-PI should be corrected to FITC-PI.
4. NF-KB should be corrected to NF-κB.
5. For CD86⁺, CD206⁺, and F4/80⁺, the "+" should be superscripted.
6. L79, 185 and 537 should be re-checked.
7. Arabic numerals cannot be used as sentence starters in English writing. Full English spelling should be used when required such as L558 and L661.
8. In L175–176, the "Co-Cy" should be correct to "C-Cy".
9. The legend of Figure 7p and 7q is not consistent with the Figure 7p and 7q.
10. Lines 611 through 613 are not written in complete sentences.

Reviewer #3

(Remarks to the Author)

In this manuscript, the authors designed a cobalt-based nanoplateform (PfCC), which integrates an antimicrobial peptide-functionalized membrane camouflage. This platform synergistically suppresses colorectal cancer (CRC) by modulating endogenous H₂S levels and leveraging photothermal effects. Additionally, it is combined with an AI algorithm to accurately quantify tumor therapeutic effects. The study yielded several noteworthy conclusions, but certain aspects remain unaddressed. To enhance the publishability, I suggest addressing the following specific points outlined below.

1. In Figure 2b, the mark MCC is missing from the middle column.
2. Figure S1 is high-resolution XPS spectra of N 1s and S 2p, please add Co 2p. The spectrum of S 2p is too weak to determine, and the intensity of the peaks is only comparable to the background noise.
3. The authors need to demonstrate successful synthesis by X-ray diffraction (XRD) characterization of the prepared materials, such as Co-MOF and the precipitated CoS.
4. In Figure 2j, the authors claim that due to the acid-responsive property of cobalt-based MOFs, the material completely degraded after 2 h at pH 6.5. Are there additional factors for this, such as the presence of ligands containing proton-sensitive groups? Furthermore, the extent of degradation can potentially be quantified using Inductively Coupled Plasma (ICP) analysis.
5. Since the photothermal effect of CoS precipitate formation can inhibit tumor growth, the photothermal conversion efficiency of CoS should be supplemented.
6. In Figure 3k, why does the addition of H₂S combined with exposure to NIR induce necrosis (Q1 region), but not apoptosis (Q2 and Q3 region)? Please provide a detailed explanation.
7. Co exhibits certain toxicity, in order to avoid long-term toxicity, how long can Co element be metabolized in mice and eliminated from the body?

Version 2:

Reviewer comments:

Reviewer #2

(Remarks to the Author)

The authors' responses generally addressed my comments, and the additional studies improved the manuscript. It is recommended to accept this manuscript after the writing has been carefully double-checked. Some minor points are listed below.

- 1.P5, L108, "Based on" should be corrected to "based on".
- 2.Please uniform the writing of "in situ".
- 3.In supporting information, an n-dash, not a hyphen, should be used to indicate a range of numbers.
- 4.In supporting information, Arabic numerals cannot be used as sentence starters in English writing. Full English spelling should be used when required.

Reviewer #3

(Remarks to the Author)

All concerns we raised have been addressed. I recommend acceptance.

In this manuscript, the authors designed a cobalt-based nanoplatfom (PfCC), which integrates an antimicrobial peptide-functionalized membrane camoufla. This platform synergistically suppresses colorectal cancer (CRC) by modulating endogenous H₂S levels and leveraging photothermal effects. Additionally, it is combined with an AI algorithm to accurately quantify tumor therapeutic effects. The study yielded several noteworthy conclusions, but certain aspects remain unaddressed. To enhance the publishability, I suggest addressing the following specific points outlined below.

1. In Figure 2b, the mark MCC is missing from the middle column.
2. Figure S1 is high-resolution XPS spectra of N 1s and S 2p, please add Co 2p. The spectrum of S 2p is too weak to determine, and the intensity of the peaks is only comparable to the background noise.
3. The authors need to demonstrate successful synthesis by X-ray diffraction (XRD) characterization of the prepared materials, such as Co-MOF and the precipitated CoS.
4. In Figure 2j, the authors claim that due to the acid-responsive property of cobalt-based MOFs, the material completely degraded after 2 h at pH 6.5. Are there additional factors for this, such as the presence of ligands containing proton-sensitive groups? Furthermore, the extent of degradation can potentially be quantified using Inductively Coupled Plasma (ICP) analysis.
5. Since the photothermal effect of CoS precipitate formation can inhibit tumor growth, the photothermal conversion efficiency of CoS should be supplemented.
6. In Figure 3k, why does the addition of H₂S combined with exposure to NIR induce necrosis (Q1 region), but not apoptosis (Q2 and Q3 region)? Please provide a detailed explanation.
7. Co exhibits certain toxicity, in order to avoid long-term toxicity, how long can Co element be metabolized in mice and eliminated from the body?

Editor:

1. Please complete or update the following checklist(s) to verify compliance with our research ethics and data reporting standards. Address all points on the checklist, revising your manuscript in response to the points if needed.

The form(s) must be downloaded and completed in Adobe Reader rather than opened in a web browser. Each form must be uploaded as a Related Manuscript file at the time of resubmission.

Editorial policy checklist:

<https://www.nature.com/documents/nr-editorial-policy-checklist.pdf>

Reporting summary:

Reply: Thanks for your suggestion. We have downloaded and completed the form(s) in Adobe Reader, and each form was uploaded as a Related Manuscript file.

2. Nature journals have recently announced an update to our guidance on reporting on sex and gender in research studies (see here). We strongly encourage researchers to follow the ‘Sex and Gender Equity in Research – SAGER – guidelines’ and to include sex and gender considerations for studies involving humans, vertebrate animals and cell lines where relevant to the topic of study (an overview can be found here). Authors should use the terms sex (biological attribute) and gender (shaped by social and cultural circumstances) carefully in order to avoid confusing both terms.

When preparing your revised manuscript, please be aware of our guidance on Sex and Gender reporting).

Please note that we require that the following recommendations from the guidelines are followed:

1. If the research findings apply to only one sex or gender, that must be indicated in the title and/or abstract.

2a. For studies involving vertebrates animal and cell lines- The Reporting Summary should include whether sex was considered in the study design.

2b. For studies involving human research participants- The Reporting Summary should include whether sex and/or gender was considered in the study design and whether sex and/or gender of participants was determined based on self-report or assigned (and methodology used).

3. Data should be reported disaggregated for sex and gender where this information has been collected and consent has been obtained for reporting and sharing individual-level data; disaggregated numbers for individual experiments must be provided in the source data as appropriate whereas overall numbers may be provided in the Nature Portfolio Reporting Summary.

Information on the points above should be included in the revised manuscript and detailed in the cover letter.

In addition, please note that if sex- and gender-based analyses have been performed a priori, results should be reported regardless of positive or negative outcome. We discourage conducting post hoc sex- and gender-based analysis if the study design is insufficient (for example, low sample size) to enable meaningful conclusions.

If no sex- and gender-based analyses have been performed, please indicate the reasons for the lack of these analyses in the Reporting Summary.

Reply: Thanks for your suggestion. The colorectal cancer research we conducted did not make any distinction based on gender.

3. Your work characterises chemical or biomolecular materials. Please see the link below for reporting requirements. There is no form to upload but you may need to revise your manuscript to comply with this policy.

<https://www.nature.com/ncomms/submit/chemical-characterisation>

Reply: Thanks for your suggestion. We have carefully read this section “characterisation of chemical and biomolecular materials”, checked and revised the full manuscript, so as to provide adequate data to support our assignment of identity, purity and homogeneity for compounds and materials described in the manuscript.

4. All Nature Communications manuscripts must include a “Data Availability” section after the Methods section but before the References. If any of the data can only be shared on request or are subject to restrictions, please specify the reasons and explain how, when, and by whom the data can be accessed. For more information on this policy and a list of examples, see:

<https://www.nature.com/documents/nr-data-availability-statements-data-citations.pdf>

Reply: Thanks for your suggestion. This section “Data Availability” was added before the References.

5. As Nature Portfolio policies strongly encourage you to share your research data in a public repository (e.g. spreadsheets, text, images), we are partnering with the figshare repository so that you can use the figshare integration via the ‘Research Data Deposition’ tab when submitting your revised manuscript.

Data are stored privately until a manuscript decision is reached and you can edit/withdraw them up to this point: you retain rights and control over your data. The data will be published at the same time as your article; you will receive a data DOI, with guidance on linking the data and manuscript. In the event your manuscript is not accepted, you can keep or remove your data in figshare.

We recommend the use of discipline-specific repositories where available and for a number of data types this is mandatory. Ensure you do not submit these data types or any sensitive data to figshare.

We strongly encourage you to deposit all new data associated with the paper in a persistent repository where they can be freely and enduringly accessed. We recommend submitting the data to discipline-specific and community-recognised repositories; a list of repositories is provided here: <https://www.nature.com/sdata/policies/repositories>

Refer to our data policies here: <https://www.nature.com/nature-portfolio/editorial-policies/reporting-standards#availability-of-data>

Reply: Thanks for your suggestion. We have uploaded the Source Data to the Figshare.

6. Proteomic datasets must be deposited in a publicly accessible database, and accession codes and associated hyperlinks must be provided in the “Data Availability” section.

Reply: Thanks for your suggestion. This manuscript did not involve proteomic datasets.

7. All DNA sequencing, RNA sequencing or microarray data must be deposited in an approved, publicly accessible repository listed here (<https://www.nature.com/nature-research/editorial-policies/reporting-standards#availability-of-data>), and relevant accession codes should be stated in the data availability statement.

Reply: Thanks for your suggestion. All RNA sequencing has been deposited in the Figshare, and relevant accession codes also have been stated in the data availability statement.

8. To maximise the reproducibility of research data, we ask that you provide a Source Data file containing the raw data underlying the following types of display items:

- Any reported means/averages in box plots, bar charts, and tables
- Dot plots/scatter plots, especially when there are overlapping points
- Line graphs
- Uncropped and unprocessed scans of all blots and gels including all quantified replicates. The edge of membranes, molecular weight ladders and loading controls should be presented on all blots. Where membranes have been cut, please ensure that at least one marker above and below is present. For an example of presentation of full scan blots, see the Source Data file of <https://www.nature.com/articles/s41467-020-16984-1#Sec35> and for more information, please refer to <https://www.nature.com/nature-research/editorial-policies/image-integrity>.

The data should be provided in a single Excel file with data for each figure/table in a separate sheet, or in multiple labelled files within a zipped folder. Name this file or folder ‘Source Data’, and include a brief description in your cover letter. The “Data Availability” section should also include the statement “Source Data are provided with this paper.”

To learn more about our motivation behind this policy, please see: <https://www.nature.com/articles/s41467-018-06012-8>

A Source Data file is not necessary if all display items presented in the main manuscript and supplementary information can be reproduced from raw data and code that have already been shared in a public repository.

Reply: Thanks for your suggestion. We have provided the Source Data file, and shared in Figshare.

9. Please replace your bar graphs with plots that feature information about the distribution of the underlying data. All data points should be shown for plots with a sample size less than 10. For larger sample sizes, please consider box-and-whisker or violin plots as alternatives. Measures of centrality, dispersion and/or error bars should be plotted and described in the figure legend.

Reply: Thanks for your suggestion. We have replaced all our bar graphs to plots that feature information about the distribution of the underlying data.

10. Nature Communications is committed to improving transparency in authorship. As part of our efforts in this direction, we are now requesting that all authors identified as ‘corresponding author’ create and link their Open Researcher and Contributor Identifier (ORCID) with their account on the Manuscript Tracking System prior to acceptance. ORCID helps the scientific community achieve unambiguous attribution of all scholarly contributions.

You can create and link your ORCID from the home page of the Manuscript Tracking System by clicking on ‘Modify my Springer Nature account’ and following these instructions. Please also inform all co-authors that they can add their ORCIDs to their accounts and that they must do so prior to acceptance.

If you experience problems in linking your ORCID, please contact the Platform Support Helpdesk.

Reply: Thanks for your suggestion. We have added the ORCID of the corresponding author and other authors who already possess an ORCID. Such as Yuan-Di Zhao: orcid.org/0000-0002-4286-4275, Jiang Xia: orcid.org/0000-0001-8112-7625, Kai Cheng: orcid.org/0000-0002-7918-3465, Bo Liu: orcid.org/0000-0002-5450-5528, Jin-Xuan Fan: orcid.org/0000-0002-8434-7591.

11. If there are any changes to the author list in the revised manuscript, please use this approval form www.nature.com/documents/nr-author-list-change-form.pdf, arranging for all authors on your paper to sign the statement confirming that they agree to the author list being changed, and add this document to your resubmission.

Reply: Thanks for your suggestion. We have added an author, Yan-Bin Guo, and all authors have agreed and signed the consent form, which is uploaded to the system.

12. Please use the link below to submit the following items as separate documents:

- Revised manuscript
- Any supplementary files
- Point-by-point response to the reviewers’ comments, reproduced verbatim
- Cover letter to the editor
- Any completed checklist(s)

Reply: Thanks for your suggestion. We have prepared the above files.

Reviewer #1 (Remarks to the Author):

This manuscript presents a biomimetic nano-platform by utilizing antimicrobial peptide-functionalized CRC cell membranes to encapsulate a cobalt-based metal-organic framework (PfCC) for the multimodal treatment of colorectal cancer (CRC). This therapeutic strategy combines photothermal therapy and immunomodulation of tumor microenvironment via activating the innate immune response and regulating the endogenous H₂S level and the balance of intestinal flora, providing a new idea for CRC treatment. The authors comprehensively validated the nanoplatform from synthetic characterization, in vitro cells and in vivo animals. In particular, the quantitative analysis of in situ CRC by AI deep learning models (e.g., SAM) provides a new approach for the assessment of tumor treatment effects. However, some problems should be resolved before considering publication.

Major issues:

1. The content of antimicrobial peptide and Co in PfCC needs to be determined.

Reply: Thanks for your suggestion. The content of antimicrobial peptide and Co in PfCC were quantified. First, 1 mL of PfCC were dried by freeze-drying, and weighed to obtain 4.75 mg of solid powder. With using an atomic absorption spectrophotometer, the standard curve for cobalt was plotted (**Figure S3a**), and the cobalt ions were quantified in PfCC. The results showed that the concentration of cobalt ions was 0.8076 mg/mL, accounting for 17.00% of the PfCC probe. BCA protein concentration analysis revealed that, after subtracting the protein content in the cell membrane, the protein content of antimicrobial peptide was 0.152 mg/mL, accounting for 3.20% of the PfCC probe (**Figure S3b**).

This section was added in line 26, page 8 and lines 1–6, page 9.

Figure S3. a The standard curve for cobalt concentration and absorption. **b** The proportion of cobalt and antimicrobial peptides in the PfCC probe.

2. The regulatory effects of antimicrobial peptide and PfCC on different bacteria need to be determined by in vitro assessment.

Reply: Thanks for your suggestion. To investigate the antibacterial effect of antimicrobial peptide (AMP) on bacteria (taking Escherichia and Vibrio as examples), different concentrations of AMP (0,

10, 50, 100, 200 $\mu\text{g}/\text{mL}$) were mixed with the bacterial cultures, and incubated for 1 h. Then, 100 μL of each bacterial suspension was spread on LB agar plates and cultured at 37 $^{\circ}\text{C}$ for 18 h. The colonies on the LB agar plates were recorded. The results showed that as the concentration of AMP increased, the growth of *Escherichia* and *Vibrio* was significantly inhibited. Moreover, even at a concentration as low as 10 $\mu\text{g}/\text{mL}$, it could still have a good antibacterial effect (**Figure S31**). Subsequently, we further explored the effects of different probes on different bacteria. The results showed that both AMP and PfCC had significant inhibitory effects on *Escherichia* and *Vibrio*, and were slightly better than penicillin (**Figure 8a and b**). This further demonstrated the great potential of antimicrobial peptides in regulating bacterial proliferation *in vivo*.

This section was added in lines 14–23, page 38.

Figure S31. The inhibitory effects of different concentrations of antimicrobial peptides on the growth of *Escherichia coli* and *Vibrio* bacteria.

Figure 8. a, b. The inhibitory effects of different probes on the growth of *Escherichia coli* and *Vibrio* bacteria.

3. *In vitro* cellular experiments, including the effect of H_2S on the photothermal effect of PfCC, staining of live and dead cells, and apoptosis assay, all of which need to include the experimental groups of H_2S only and H_2S + PfCC without NIR as controls.

Reply: Thanks for your suggestion. For the effect of H_2S on the photothermal effect of PfCC, we continued to conduct infrared thermal imaging studies on different treatment groups, and it was found

that the temperature of the PBS, H₂S, and H₂S+PfCC groups remained basically unchanged without NIR irradiation. After adding 2.1 mM of H₂S to PfCC and irradiating with NIR laser, the temperature could change by about 20 °C within 5 min (**Figure S6**).

This section was added in lines 4–8, page 12.

To explore the effect of H₂S on the photothermal effect of the probe, we added H₂S and H₂S + PfCC without NIR groups as controls. The results showed that with increasing H₂S concentration, the survival rate of CT26 cells gradually decreased (**Figure 3e**), which was significantly different from that of the control group ($p < 0.001$).

This section was added in lines 7–8, page 15.

For the staining of live and dead cells, we added H₂S and H₂S + PfCC without NIR groups as controls. The results showed that the H₂S alone and the H₂S + PfCC group both exhibited green fluorescence, indicating that the apoptotic effect on cells caused by low concentrations of H₂S and the simple CoS precipitate was relatively small (**Figure 3j**).

This section was added in lines 4–6, page 17.

For the apoptosis assay, we also added H₂S and H₂S + PfCC without NIR groups as controls. The results showed when simple H₂S was added (**Figure 3k**), only a few late-stage apoptotic cells (2.4%) were present; after adding PfCC, late-stage apoptotic cells changed less, which indicated that the simple H₂S and formation of CoS precipitates generally didn't cause any damage to the CT26 cells.

This section was added in lines 20–22, page 17.

Figure S6. Temperature change of different treatments followed with/without laser treatment at 1.0 W/cm².

Figure 3. e Survival rates of CT26 cells after incubation with PfCC treated with different concentrations of H₂S standard solution and irradiation with NIR, ***: $p < 0.001$. j, k Calcein/PI staining and flow cytometric analysis of apoptosis in CT26 cells incubated with different concentrations of H₂S-treated MCC.

4. In in vitro anti-tumor assay, the quantitative analysis data of macrophages and NK cells should be provided to demonstrate the regulation effect of PfCC on the innate immune response.

Reply: Thanks for your suggestion. We conducted the quantitative analysis data of macrophages and NK cells in tumours, spleen, and lymph nodes. The results showed compared with the PBS group, the distribution of F4/80⁺ macrophages of the tumour and lymph in the MCC group gradually increased; compared with MCC+NIR group, the macrophages of the tumour and lymph in the PfCC+NIR treatment group were also significantly increased, indicating that MCC and antimicrobial peptides could induce macrophages response in tumours (Figure 7m).

This section was added in lines 3–9, page 36.

We also performed analysis of different macrophage types within the tumour, spleen, and lymph nodes. The results showed that after MCC treatment, the proportion of M1 macrophages in the tumour increased compared with the PBS group, and after treatment with PfCC+NIR group, the proportion of M1 macrophages in the tumour further increased, while M2 macrophages decreased, this was because low concentrations of H₂S can promote the polarization of M2 to M1 macrophages. And the change in the lymph nodes was just the opposite, which further illustrated that the PfCC may further enhance the innate immune response in the tumour by promoting the migration of M1 macrophages in the lymph nodes into the tumour (Figure S29a).

Finally, we conducted the analysis of NK cells within the tumour, spleen, and lymph nodes. The results

showed that the amount of NK cells in the tumour increased with the treatment of MCC, and further increased after further treatment with the released peptides in PfCC, and appeared decreasing trend in the spleen and lymph (Figure 7n and S29b). This also demonstrated that PfCC could affect the distribution of NK cells in the tumour and had the preliminary effect of activating the innate immune response in the tumour.

This section was added in lines 24–26, page 36, and line1, page 37.

Figure 7m. Flow cytometry quantification of the percentage of F4/80+ macrophage in tumours, spleen, and lymph nodes. Data are presented as the means±S.D. (n=3 independent samples).

Figure S29. Flow cytometry quantification of M1 macrophages/M2 macrophages, and NK cells in tumours, spleen, and lymph nodes.

5. Details of AI algorithms (e.g., model parameters, training data) need to be supplemented to enhance the reproducibility of results.

Reply: Thanks for your suggestion. Firstly, we built a deep learning environment based on Python, which was mainly divided into the 5 parts. 1). Installed Pycharm and Anaconda3. 2) Created a virtual environment in Anaconda3 and installed Python: after installing Anaconda, we logged in the Anaconda Prompt, Conda create -n environment name and python = 3.10, then continuing running. 3). Installed Pytorch, that was, entering Conda activate environment name in the anaconda prompt, entering the previously created virtual environment and the Pytorch official website command to run. 4). Opened the project with Pycharm, configured the virtual environment, and installed the corresponding Opencv, Matplotlib, Scipy, segment_anything and other dependent packages in our created environment based

on the Anaconda Prompt. 5). There were three models (h, l, b) in the demo.py file, among which the h model requires the largest video memory and has the best segmentation effect, and the b model requires the smallest video memory and has the worst segmentation effect. Select the corresponding model to run.

After building the deep learning environment, we started training the data. 38 groups (**Figure S26a-c**) of tumours were photographed endoscopically, and the actual tumour size was determined by dissecting the intestines of mice. Through deep machine learning, the visualization interface was designed based on the open source large-scale model segment anything model (SAM), with the lesion area automatically segmented according to the model prompts. After generating the lesion area mask, the convex hull algorithm was to calculate the convex hull of the tumour region and extract the boundary pixel coordinates, with traversing to calculate the maximum pixel distance of the region. We recorded the pixel distance automatically calculated by the algorithm and the actual measured distance of the anatomical intestine, the pixel distance obtained by the algorithm was used as the X-axis and the actual distance as the Y-axis, and the linear regression model was fitted to obtain the standard curve.

This section was added in supporting information, page 13-14, and the results were added as S26.

Figure S26. a-c Original tumour, segmented tumour, extracted tumour images based on large model training.

6. The discussion section is only a brief conclusion. It is recommended to add the potential challenges of PfCC in clinical translational research (e.g., feasibility of large-scale production, long-term safety *in vivo*) as well as comparisons with other H₂S-modulated or photothermal therapies, highlighting the significance of this study.

Reply: Thanks for your suggestion. The discussion section has been modified as follows:

This work successfully synthesized a biomimetic nano-platform (PfCC) leveraging peptide modified-CRC cell membrane coating and a degradable cobalt-based MOF for synergistic *in situ* CRC therapy.

PfCC achieved tumor-targeted enrichment through homologous homing effect, where the acidic degradation triggered three coordinated anti-tumor mechanisms: 1). H₂S scavenging: released Co²⁺ consumed endogenous H₂S via precipitation (CoS). 2). Downregulation of H₂S production: the released peptides modulated the tumour microbiome such as *Desulfovibrio*, sustaining low H₂S levels. 3). Immunomodulation & PTT: reduced H₂S inhibited pro-tumour transcription factors, promoting M2-to-M1 macrophage polarization. Concurrently, CoS precipitate formation enabled PTT, further reversing immunosuppression and activating innate immunity.

While PfCC demonstrated compelling efficacy in preclinical models, its path to clinical application faces key challenges. Reproducible large-scale production of cell-membrane vesicles with consistent peptide insertion efficiency and coating homogeneity requires robust GMP-compliant processes. Additionally, standardizing extrusion methods and ensuring batch-to-batch consistency are critical hurdles. For the long-term safety, as the above study, the 7-day metabolic rate of cobalt ions in the body basically reached 90%, indicating that there were no potential long-term toxicity.

Here, PfCC offered distinct advantages over existing H₂S-modulating or photothermal strategies. Compared with conventional H₂S Inhibitors such as small-molecule H₂S inhibitors (e.g., AOAA, PAG), they often suffer from poor tumour specificity, systemic toxicity, and inability to address both H₂S overproduction and excess existing H₂S. But PfCC uniquely combined simultaneous H₂S scavenging and sustained production blockade (peptide/microbiome modulation) within the tumour microenvironment (TME), enhancing efficacy and potentially reducing off-target effects. For traditional PTT Agents, while inorganic PTT agents such as Au nanorods and CuS nanoparticles are potent, their non-degradability raises long-term safety concerns. Whereas CoS precipitate provided localized PTT only upon TME-specific degradation, minimizing background toxicity. Furthermore, the PTT effect was not the sole mechanism but synergized with immunomodulation triggered by H₂S depletion.

Beyond the therapeutic platform, this study pioneered the use of AI large model learning to assist in developing convex hull algorithms for the quantitative assessment of *in situ* CRC tumours. This approach will provide a robust, objective tool for evaluating therapeutic response *in vivo* and establishes a valuable algorithmic framework adaptable to other *in situ* tumour models.

This section was added in page 41, and lines 1–5, page 42.

Minor issues:

1. English writing should try to use short sentences to enhance readability. Some sentences of this manuscript are long and it is recommended to simplify them.

Reply: Thanks for your suggestion. We have simplified some long sentences of this manuscript, and the revised sentences were marked with Red.

2. An n-dash not hyphen should be used to indicate a range of numbers. For example, “2-6” should be correct to “2–6”.

Reply: Thanks for your suggestion. We have revised connection symbol between a range of numbers in the literature, as well as the full manuscript.

3. FTTC-PI should be corrected to FITC-PI.

Reply: Thanks for your suggestion. We are sorry for such a mistake, and have revised and checked the full manuscript.

4. NF-KB should be correct to NF-κB.

Reply: Thanks for your suggestion. We have corrected and carefully checked the full manuscript.

5. For CD86⁺, CD206⁺, and F4/80⁺, the “+” should be superscripted.

Reply: Thanks for your suggestion. We have corrected the “+” to be superscripted, and carefully checked the full manuscript.

6. L79, 185 and 537 should be re-checked.

Reply: Thanks for your suggestion. We have revised the L79, 185 and 537, and checked the full manuscript.

7. Arabic numerals cannot be used as sentence starters in English writing. Full English spelling should be used when required such as L558 and L661.

Reply: Thanks for your suggestion. We have revised the arabic numerals of the sentence starters to full English spelling, and checked the full manuscript.

8. In L175–176, the “Co-Cy” should be correct to “C-Cy”.

Reply: Thanks for your suggestion. We have revised “Co-Cy” to “C-Cy”, and checked the full manuscript.

9. The legend of Figure 7p and 7q is not consistent with the Figure 7p and 7q.

Reply: Thanks for your suggestion. We have revised the legend of Figure 7p and 7q.

10. Lines 611 through 613 are not written in complete sentences.

Reply: Thanks for your suggestion. Lines 611 through 613 has been revised as “The main method involved preparing a standard curve, followed by grinding and digesting tumour tissue, adding the WSP-1 probes, measuring the fluorescence signal after a specific incubation period, and calculating the H₂S concentration via the standard curve.”

Reviewer #2 (Remarks to the Author):

In this manuscript, the authors designed a cobalt-based nanoplatform (PfCC), which integrates an antimicrobial peptide-functionalized membrane camouflage. This platform synergistically suppresses colorectal cancer (CRC) by modulating endogenous H₂S levels and leveraging photothermal effects. Additionally, it is combined with an AI algorithm to accurately quantify tumor therapeutic effects. The study yielded several noteworthy conclusions, but certain aspects remain unaddressed. To enhance the publishability, I suggest addressing the following specific points outlined below.

1. In Figure 2b, the mark MCC is missing from the middle column.

Reply: Thanks for your suggestion. We are sorry for such a mistake, and have corrected and carefully checked the full manuscript.

2. Figure S1 is high-resolution XPS spectra of N 1s and S 2p, please add Co 2p. The spectrum of S 2p is too weak to determine, and the intensity of the peaks is only comparable to the background noise.

Reply: Thanks for your suggestion. We have re-conducted high-resolution XPS detection of S 2p, and added N 1s, S 2p, and Co 2p to Figure S1. Here, Co 2p 3/2 and Co 2p1/2 were located at 781.08 eV and 798.58 eV, N 1s at 400.83 eV, S 2p at 165.63 eV.

This section was added in lines 12, page 8.

Figure S1. a-c High-resolution X-ray photoelectron spectra of the elements Co, N, and S in PfCC probe.

3. The authors need to demonstrate successful synthesis by X-ray diffraction (XRD) characterization of the prepared materials, such as Co-MOF and the precipitated CoS.

Reply: Thanks for your suggestion. We have performed XRD characterization of Co-MOF and the generated CoS. The results (Figure S5a) showed that the obvious diffraction peaks of Co-MOF at $2\theta = 7.23^\circ, 10.38^\circ, 12.7^\circ, 14.78^\circ, 16.5^\circ, 18.08^\circ, 22.13^\circ, 24.39^\circ,$ and 26.69° correspond to the crystal faces (011), (002), (112), (022), (013), (222), (114), (233), and (044), of ZIF-67(PDF#43–0144). In the case of CoS (Figure S5b), the relatively intense peaks at $2\theta = 31.59^\circ, 35.52^\circ, 44.36^\circ,$ and 55.31° correspond to the (002), (020), (200), and (120) lattice planes of the structure of CoS respectively; this observation is consistent with the standard card for CoS(PDF#65–3418).

This section was added in lines 19–25, page 11.

Figure S5. a, b X-ray powder diffraction analysis of Co-MOF and CoS.

4. In Figure 2j, the authors claim that due to the acid-responsive property of cobalt-based MOFs, the material completely degraded after 2 h at pH 6.5. Are there additional factors for this, such as the presence of ligands containing proton-sensitive groups? Furthermore, the extent of degradation can potentially be quantified using Inductively Coupled Plasma (ICP) analysis.

Reply: Thanks for your suggestion. The presence of ligands containing proton-sensitive groups is indeed the main factor influencing the acid response of MOFs. The stability of MOF is highly dependent on the coordination bonds between the organic ligands and the metal ions/clusters. During the synthesis of cobalt-based MOFs, we synthesized a stable framework structure through the coordination reaction between cobalt nitrate and dimethylimidazole. However, the main ligand groups such as carboxylic acid, nitrogen-containing heterocyclic, amino, and other oxygen-containing groups are prone to be affected by protonation under acidic conditions. The N atom on the dimethylimidazole ring is basic, and can accept protons to form imidazolium ions, which will destroy its coordination ability with Co. After the cobalt ions are protonated by the ligands, they become unstable, and ultimately lead to the collapse of the framework.

Four aliquots of 1 mg/mL MCC were placed in PBS at pH 6.5 for 0.1, 0.5, 1, and 2 h, and after centrifugation, the supernatant were collected for detection. With the quantitative analysis using Inductively Coupled Plasma (ICP), the results showed that after 0.1 h (**Figure S4**), the concentration of cobalt ions in the supernatant was 17.8 $\mu\text{g/mL}$, and 2 h later, the concentration reached 151.2 $\mu\text{g/mL}$, which showed a significant difference compared to that of 0.1 h, meaning the complete degradation of Co-MOF in MCC.

This section was added in lines 12–17, page 11.

Figure S4. Quantitative analysis of Co during the degradation of PfCC at different times under pH=6.5. **: $p < 0.01$.

5. Since the photothermal effect of CoS precipitate formation can inhibit tumor growth, the photothermal conversion efficiency of CoS should be supplemented.

Reply: Thanks for your suggestion. We have conducted the photothermal conversion efficiency testing experiments of CoS (**Figure S12**). Since CoS is poorly soluble in water, 1 mg of CoS powder was added to 1 mL of DMSO containing 0.01 mg of sodium dodecyl sulfate. After ultrasonication for 30 mins, the Uv-vis absorption spectrum absorption of CoS at 808 nm was measured as 1.31. 300 µL of CoS was placed in a quartz cuvette and irradiated with an 808 nm laser for 5 mins, then recording the temperature of heating and cooling processes. According to the formula, $\eta = \frac{h_s(T_{Max}-T_{Surr})-Q_{Dis}}{I(1-10^{-A_{808}})}$, $T_{max}=53.5$, $T_{surr}=26.1$, $A_{808}=1.31$, the power intensity is 600 mW, and $h=mc/\tau$, $t=\tau (-\ln\Theta)$, the photothermal conversion efficiency could be calculated to be 20.72%.

This section was added in lines 21–26, page 15, and lines 1–2, page 16.

Figure S12. a, b Rise-fall temperature of CoS and calculation of τ value chart.

6. In Figure 3k, why does the addition of H₂S combined with exposure to NIR induce necrosis (Q1 region), but not apoptosis (Q2 and Q3 region)? Please provide a detailed explanation.

Reply: Thanks for your suggestion. Due to the addition of H₂S, the precipitation reaction of cobalt ions was induced. Under the NIR irradiation, CoS had a relatively good ability to kill cells through

photothermal effect. With the differences in irradiation time and the formation of precipitates, a strong photothermal reaction may induce excessive late apoptosis (Q2) and necrosis (Q1). When low concentration of H₂S was added, the mild photothermal heating induced by H₂S was not enough to cause large-area necrosis of cells (Q1), but more early apoptosis (Q3) and late apoptosis (Q2). As the concentration of H₂S increased, the degree of late apoptosis became too severe, the probability of necrosis may gradually increase, which also showed that continuous local hyperthermia will cause increased cell necrosis. From the Figure 3k, when CT26 cells incubated with MCC were added with 20 µg/mL of H₂S and treated with laser irradiation, the early and late apoptosis were 27.0 and 14.5%, respectively, and the cell necrosis reached 23%. But when the H₂S concentration was 80 µg/mL, the late-stage apoptosis and necrosis reached 27.1% and 27.6%, and the total percentages of the early stage apoptosis, later stage apoptosis and necrosis were 64.5%, 68.9%, and 74.1% respectively, indicating that high concentrations of H₂S may induce a stronger photothermal effect of Co²⁺ generated by acid degradation in CT26 cells.

This section was added in lines 21–26, page 17, and lines 1–6, page 18.

7. Co exhibits certain toxicity, in order to avoid long-term toxicity, how long can Co element be metabolized in mice and eliminated from the body?

Reply: Thanks for your suggestion. An *in vivo* metabolism study of PfCC probe was conducted in CT26 tumor-bearing mice at different time. The results (**Figure S18a and b**) showed that after 24 h injection of PfCC probe, the metabolic elimination rate of cobalt ions in the blood reached 99.9%; 99.8% in the heart; 27.1% in the liver, and 91.5% after 7 d; 45.3% in the spleen, and 89.0% after 7 d; 87.9% in the lung, and 99.9% after 7 d; 88.0% in the kidney, and 100% after 7 d; 31.3% in the tumor, and 97.7% after 7 d. In addition, the long-term blood routine, blood biochemistry and tissue section results also showed (**Figure S17**) that during the 20 to 45 d, the blood routine and blood biochemistry indicators of the mice did not change significantly compared with the PBS group, and there were no obvious pathological changes in the heart, liver, spleen, lung, and kidney. All the above results further illustrated that the cobalt ions released by the PfCC probe had neglectable effect on the body.

This section was added in lines 22–26, page 19.

Figure S17. Biosafety of the probe *in vivo*. Blood analysis of normal mice after Co@PM injection: **a-d** WBC, RBC, HGB, and PLT. **e, f** Analysis of liver function indicators: ALT and AST. **g-k** Corresponding organ index of heart, liver, spleen, lung, and kidney; **l** the change in body weight of mice injected with PBS and probes over 45 d. **m** The HE staining results of the corresponding organs after 45 d of PBS injection, 1, 20, and 45 d of probe.

Figure S18. a, b Cobalt ion metabolism of blood, major organs and tumours. The data are presented as the mean \pm standard deviation ($n = 3$ independent samples).

Reviewer #2 (Remarks to the Author):

The authors' responses generally addressed my comments, and the additional studies improved the manuscript. It is recommended to accept this manuscript after the writing has been carefully double-checked. Some minor points are listed below.

1. P5, L108, “Based on” should be corrected to “based on”.

Reply: Thanks for your suggestion. We have corrected “Based on” to “based on”.

2. Please uniform the writing of “in situ”.

Reply: Thanks for your suggestion. We have made all the "in situ" items uniformly non-italicized.

3. In supporting information, an n-dash, not a hyphen, should be used to indicate a range of numbers.

Reply: Thanks for your suggestion. We have carefully checked the number range, and have changed it to n-dash in supporting information.

4. In supporting information, Arabic numerals cannot be used as sentence starters in English writing. Full English spelling should be used when required.

Reply: Thanks for your suggestion. We have revised the arabic numerals of the sentence starters to full English spelling, and checked the supporting information.

Reviewer #3 (Remarks to the Author):

All concerns we raised have been addressed. I recommend acceptance.

Reply: Thanks for your suggestion.